# A graph-based genome and pan-genome variation of the model plant *Setaria*

Qiang He [1,20], Sha Tang [1,20], Hui Zhi[1,20], Jinfeng Chen [2,20], Jun Zhang[1], Hongkai Liang[1], Ornob Alam[3], Hongbo Li [4], Hui Zhang[1,5], Lihe Xing[1], Xukai Li [6], Wei Zhang[1], Hailong Wang[1], Junpeng Shi[7], Huilong Du[8], Hongpo Wu[1], Liwei Wang[1], Ping Yang [1], Lu Xing[9], Hongshan Yan[9], Zhongqiang Song[9], Jinrong Liu[9], Haigang Wang[10], Xiang Tian[10], Zhijun Qiao[10], Guojun Feng[11], Ruifeng Guo[12], Wenjuan Zhu[12], Yuemei Ren[12], Hongbo Hao[13], Mingzhe Li[13], Aiying Zhang[14], Erhu Guo[14], Feng Yan[15], Qingquan Li[15], Yanli Liu[16], Bohong Tian[16], Xiaoqin Zhao[17], Ruiling Jia[17], Baili Feng[5], Jiewei Zhang[18], Jianhua Wei[18], Jinsheng Lai [7], Guanqing Jia [1] ✉, Michael Purugganan [3,19] ✉ & Xianmin Diao [1] ✉

*Setaria italica* (foxtail millet), a founder crop of East Asian agriculture, is a model plant for C4 photosynthesis and developing approaches to adaptive breeding across multiple climates. Here we established the *Setaria* pan-genome by assembling 110 representative genomes from a worldwide collection. The pan-genome is composed of 73,528 gene families, of which 23.8%, 42.9%, 29.4% and 3.9% are core, soft core, dispensable and private genes, respectively; 202,884 nonredundant structural variants were also detected. The characterization of pan-genomic variants suggests their importance during foxtail millet domestication and improvement, as exemplified by the identification of the yield gene *SiGW3*, where a 366-bp presence/absence promoter variant accompanies gene expression variation. We developed a graph-based genome and performed large-scale genetic studies for 68 traits across 13 environments, identifying potential genes for millet improvement at different geographic sites. These can be used in marker-assisted breeding, genomic selection and genome editing to accelerate crop improvement under different climatic conditions.

Foxtail millet (*Setaria italica*), one of the oldest domesticated grain crops in the world, is considered to have provided the foundation for the formation of early Chinese civilization. Recent archeological evidence suggests that this species was domesticated starting ~11,000 years ago from its progenitor, green foxtail (*Setaria viridis*)[1], making it contemporaneous with barley and wheat in the early agricultural transitions of human Neolithic societies. Foxtail millet is the only present-day crop species in the genus *Setaria* and has excellent drought and low soil-nutrient tolerance. Since its domestication, foxtail millet has spread across Eurasia and Africa, and more recently to the Americas, and grows in temperate, tropical and arid environments.

Critically, *Setaria* species employ C4 photosynthesis. C4 plants, which aside from foxtail millet include maize, sorghum, sugarcane and switchgrass, possess high photosynthetic efficiency and environmental adaptability, thereby maintaining critical roles in global agricultural grain and biofuel production[2,3]. However, the complexity of most C4 crop plant genomes and the lack of high-efficiency transformation systems in these species have hindered fundamental studies and

breeding in these crops. In this regard, foxtail millet and green foxtail are ideal model systems for C4 photosynthetic crop plants due to their compact diploid genomes (~420 Mb), short life cycles (~70 d) and highly efficient transformation systems[4,5]. Despite the favorable features of foxtail millet as a C4 photosynthetic model crop, which may prove pivotal in ensuring global food security[6], relatively less is known about its genomic diversity and potential for genetic improvement.

Recently, pan-genome studies in rice[7,8], soybean[9], wheat[10], barley[11], tomato[12] and potato[13] indicate that structural variants (SVs) have critical roles in crop domestication as well as trait determination[14] and genetic improvement. To date, two draft genomes[5,15] and three relatively high-quality genomes[16-18] of green foxtail and foxtail millet have been released. Coupled with population-scale short-read sequencing data, previous studies have revealed population structure in foxtail millet and green foxtail, as well as the genetic basis of several key agronomic traits[16,19-21]. However, the full spectrum of genetic variants that underlie *Setaria* domestication and its broad ecological adaptability, including the role of pan-genomic diversity, remains largely unknown.

Here we de novo assembled 110 reference-grade genomes for 35 wild, 40 landrace, and 35 modern cultivated *Setaria* accessions, and examined genome evolution in the context of foxtail millet domestication and improvement. By incorporating the foxtail millet pan-genome, we constructed the first graph-based genome sequence of *Setaria* across these multiple accessions and performed large-scale genetic studies across 13 different environments, which could serve as a foundation for foxtail millet research and breeding, providing an example for 'breeding by design' in other crops (Supplementary Fig. 1).

## Results

### Variation and evolution in *Setaria*

We collected genome-wide resequencing data for 630 wild (*S. viridis*), 829 landrace and 385 modern cultivated accessions from the *Setaria* genus with an average sequencing depth of ~15×, of which 1,004 were newly generated and 840 were from previous studies[16,21] (Supplementary Table 1). After aligning reads to the foxtail millet 'Yugu1' reference genome, we identified ~60 million single-nucleotide polymorphisms (SNPs) and 6.7 million insertions/deletions (indels) in the 1,844 accessions (Supplementary Table 2).

We performed phylogenetic and population structure analyses using 4,934,413 high-quality SNPs (minor allele frequencies ≥ 0.05 and missing genotype rates < 0.1; Fig. 1a,b and Supplementary Fig. 2a). Based on population structure analysis, we classified the wild species into four subgroups—W1, W2, W3 and W4—which are consistent with 'Central', 'Central-East', 'Central-North' and 'West-Coast' populations, respectively, in a previous study[16]. W1 is the closest population subgroup to cultivated foxtail millet, which contains all our collected Chinese green foxtail; this indicates that W1 is the wild progenitor for all cultivated foxtail millet, and is consistent with China being the domestication center for this crop (Fig. 1a).

In our previous study, cultivated foxtail millet was classified into two divergent subgroups, which are closely related to geographic/climatic distribution and farming habits[19]. Here our larger global dataset was able to further divide foxtail millet into three (C1–C3) genetically differentiated subpopulations (Fig. 1). Both TREEMIX[22] and Admixtools[23] show that the first evolutionary split is between C3 and C1/C2 subgroups, with the latter two diverging later (Supplementary Fig. 2). C1 (343 accessions) and C2 (478 accessions) were roughly consistent with type 1 and type 2 foxtail millets in the previous study[19], with the C1 population distributed in high latitudes, and C2 at relatively lower latitudes with warmer climates. The new population subgroup we identified—C3 (82 accessions)—is broadly distributed worldwide, which suggests that C3 may have better adaptation to a wider range of climates than the other two subgroups (Fig. 1c and Supplementary Fig. 3b).

### De novo assembly of 110 wild and cultivated *Setaria*

To capture the full spectrum of genetic diversity of *Setaria* which may be overlooked by short-read resequencing approaches, we de novo assembled 110 representative *Setaria* accessions, including 35 wild, 40 landrace and 35 modern cultivated accessions (Fig. 2a). We selected these accessions based on phylogenetic relationships and geographic distribution, breeding and/or research utility and subgroup distribution to ensure they are representative of genetic diversity within foxtail millet and green foxtail (Fig. 2a,b and Supplementary Notes 1–5). The accessions we selected also span phenotypic diversity and represent the continuum of phenotypes associated with domestication and improvement (Fig. 2c,d).

Three representative accessions—Me34V (wild), Ci846 (landrace) and Yugu18 (modern cultivar)—were further selected to build high-quality reference genome assemblies for *Setaria*. We de novo assembled the three genomes with CANU[24] and HERA[25] using ~110× PacBio reads and polished the assemblies using ~65× Illumina reads and corrected them with BioNano physical maps. These three genome assemblies have greater contiguity than currently available reference genomes[5,16,18], with a mean contig N50 length of >20 Mb and LTR assembly index (LAI) exceeding 20. Over 99% of Illumina short reads and 97% of embryophyte BUSCO genes could be properly mapped, suggesting high completeness. *K*-mer-based analysis also showed that all assemblies have high completeness (99.56% ± 0.04%) and quality (40.81 ± 0.52), and low false duplications (0.52 ± 0.13) (Supplementary Table 6).

For the remaining 107 accessions, we generated ~4.1 of TB PacBio long reads and ~2.2 of TB Illumina reads with average sequencing depths of around 91.1× and 48.1×, respectively (Supplementary Table 5). Average assembly contig N50 length ranged from 126.9 kb to 5.5 Mb (Supplementary Table 6), and a mean of 99.8% of Illumina short reads and 94.5% of embryophyte BUSCO genes were aligned to these assemblies (Supplementary Table 6). *K*-mer-based analysis showed that the assembled genome quality of cultivated accessions (completeness, 97.59% ± 2.02%; QV, 39.36 ± 1.78; duplication, 2.55% ± 1.16%) is higher than that of wild accessions (completeness, 91.34% ± 6.05%; QV, 30.52 ± 6.89; duplication, 4.34% ± 2.48%). Assessing genome assembly quality using long-terminal repeat retrotransposons (LTR-RTs) indicated that all 107 assemblies reached the 'reference' level (LAI > 10), of which 17 reached the 'gold standard' level (LAI > 20; Supplementary Table 6).

A total of 161.8 Mb to 199.9 Mb (46.2% ± 0.01%) of assembled sequences were annotated as transposable elements (TEs; Supplementary Table 6), with LTR/*Gypsy* and LTR/*Copia* being the two most abundant TE superfamilies. We predicted 39,907 ± 1,056 protein-coding genes in the assembled genomes, with a BUSCO score of 94.0% ± 1.7% (Supplementary Table 6), and 98.7% ± 0.075% of genes anchored on nine chromosomes. An average of 65% of exons of predicted genes were supported by transcriptome sequencing data, and 55.4% ± 1.6% of predicted genes were assigned functional terms (Supplementary Table 6).

### Pan-genomic variation in *Setaria*

We constructed the pan-genome of foxtail millet using protein-coding genes, integrating data from 80 cultivated accessions with the 28 wild accessions from the W1 subgroup (the wild progenitor), plus three previously released genomes—Yugu1 (ref. 5), xiaomi[18] and A10 (ref. 16; Supplementary Table 5). The number of gene families increased as additional genomes were added to the analysis and approached a plateau with *n* = 30 accessions (Fig. 3a). The pan-genome was composed of 73,528 gene families, of which 23.8% were core genes, 42.9% were soft core genes (present in >90% of individuals, 100–110 accessions), 29.4% were dispensable genes (present in 2–99 accessions) and 3.9% were private genes (Fig. 3a). We identified an additional 14,283 gene families in the pan-genome that are absent in the Yugu1 reference genome.

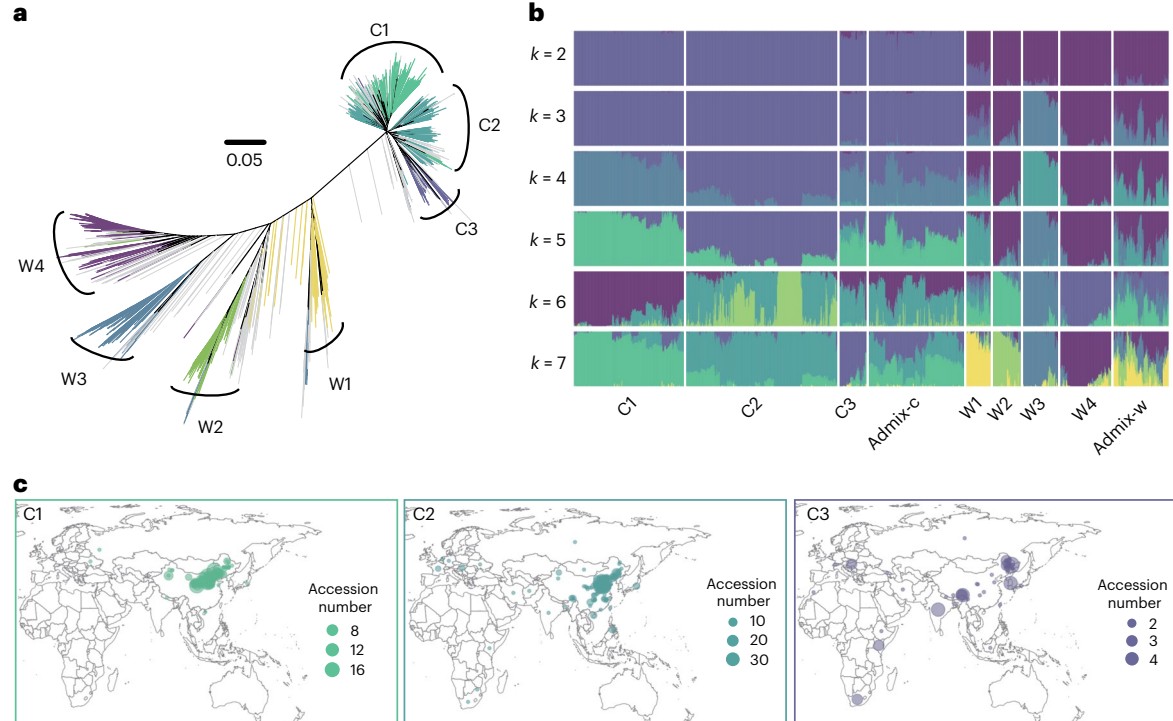

**Fig. 1 | Population structure of *Setaria*. a**, Phylogenetic tree of the 1,844 *Setaria* accessions. Gray lines indicate admixture, and other lines with different colors are subgroups corresponding to *k* = 7 in **b. b**, ADMIXTURE analysis from *k* = 2–7.

**c**, Geographic distribution of three subgroups of foxtail millet accessions. C3 is distributed broadly compared to the other two subgroups. The map was created using the map data function in the R package ggplot2.

These genes were enriched in RNA capping, light response and specific metabolic processes, such as cellular aldehyde metabolic and protein metabolic processes (Supplementary Table 7).

By leveraging the high-quality genome assemblies, we performed pair-wise genome alignment with 'Yugu1' and identified 24.3 million SNPs and 3.8 million indels (<50 bp) in the 112 accessions, 1.5% of which are nonsynonymous and may impact gene function (Supplementary Tables 8 and 9). A total of 202,884 nonredundant SVs (≥50 bp in size), comprising 107,151 insertions, 76,915 deletions, 18,455 translocations and 363 inversions, were detected (Fig. 3b and Supplementary Table 8); approximately 90% of these were shorter than 8.8 kb, 6.6 kb, 62.6 kb and 137.4 kb, respectively (Supplementary Fig. 4a). Presence–absence variants (PAVs; large insertions and deletions) are key features of crop pan-genomes, and they were the most abundant SV type (Fig. 3b and Supplementary Table 8) and tended to be enriched in intergenic repetitive regions (Fig. 3c and Supplementary Fig. 4b).

We find that most presence (72.3%; *n* = 59,429) and absence (92.8%; *n* = 99,477) variants overlapped with TEs, which are significantly higher than the proportion of TEs genome wide (60.5%; *P* < 0.001; Supplementary Fig. 4c). These TE-associated PAVs were clustered in DNA transposon regions, and most breakpoints of these PAVs were close to TE junction sites (Supplementary Fig. 4d,e), suggesting that DNA transposons may have driven the formation of most PAVs in the *Setaria* genome. We also identified 15,758 high-confidence TE-derived PAVs, which colocated with single intact TEs coupled with target site duplications (TSD).

We further analyzed the distribution of SVs based on distance from genic regions. We find, for example, that PAV numbers gradually declined as distance increased from the closest gene (Fig. 3d). We found a set of SVs localized within promoters or gene bodies of functionally significant loci, and SVs occur more frequently in genes with low expression level (Supplementary Notes 1–5 and Supplementary Figs. 5 and 6).

## SVs in foxtail millet domestication and improvement

We performed phylogenetic analysis using SVs, which clearly differentiated the 112 accessions into two distinct groups, in concordance with the SNP-based phylogeny, suggesting that SVs are also associated with *Setaria* domestication and improvement (Supplementary Fig. 7). The significant correlation of PAV density and differentially expressed genes between various population groups (two-tailed student's *t*-test, *P* = 2.2 × 10⁻¹⁶) suggest that PAVs underlie gene expression differences between populations, further strengthening the possibility that PAVs had a role in crop domestication and improvement (Supplementary Notes 1–5 and Supplementary Fig. 6).

To identify PAVs under selection during crop domestication or improvement in foxtail millet, we compared PAV frequencies between wild and landrace accessions to identify putative 'domestication' PAVs (Fig. 4a–c), and between landrace and cultivars for possible 'improvement' PAVs (Fig. 4a and Supplementary Fig. 8). We defined PAVs with substantially different frequencies between wild and landrace, and landrace and cultivars as domestication-selected SVs (domPAVs) and improvement-selected SVs (impPAVs), respectively. A total of 4,582 domPAVs (Fig. 4a–c and Supplementary Table 10) and 152 impPAVs were identified (Fig. 4a, Supplementary Fig. 8 and Supplementary Table 11), suggesting stronger selection pressure during domestication of foxtail millet compared to subsequent crop improvement. Among them, 1,933 domPAVs and 57 impPAVs are favorable PAVs (favPAVs) that have consistently elevated or reduced frequencies in both landrace and cultivated accessions. We identified 680 favorable genes that have favPAVs at the gene or promoter regions, and are enriched in biological processes related to crop domestication such as reproductive process, photoperiodism, pigment accumulation and nitrogen utilization (Fig. 4d). We also looked for colocalization between genomic regions under selection in different branches of the population tree (Supplementary Fig. 3) and these selected PAVs; we find that ten of these selected regions overlap with domPAVs and impPAVs (Supplementary Table 4).

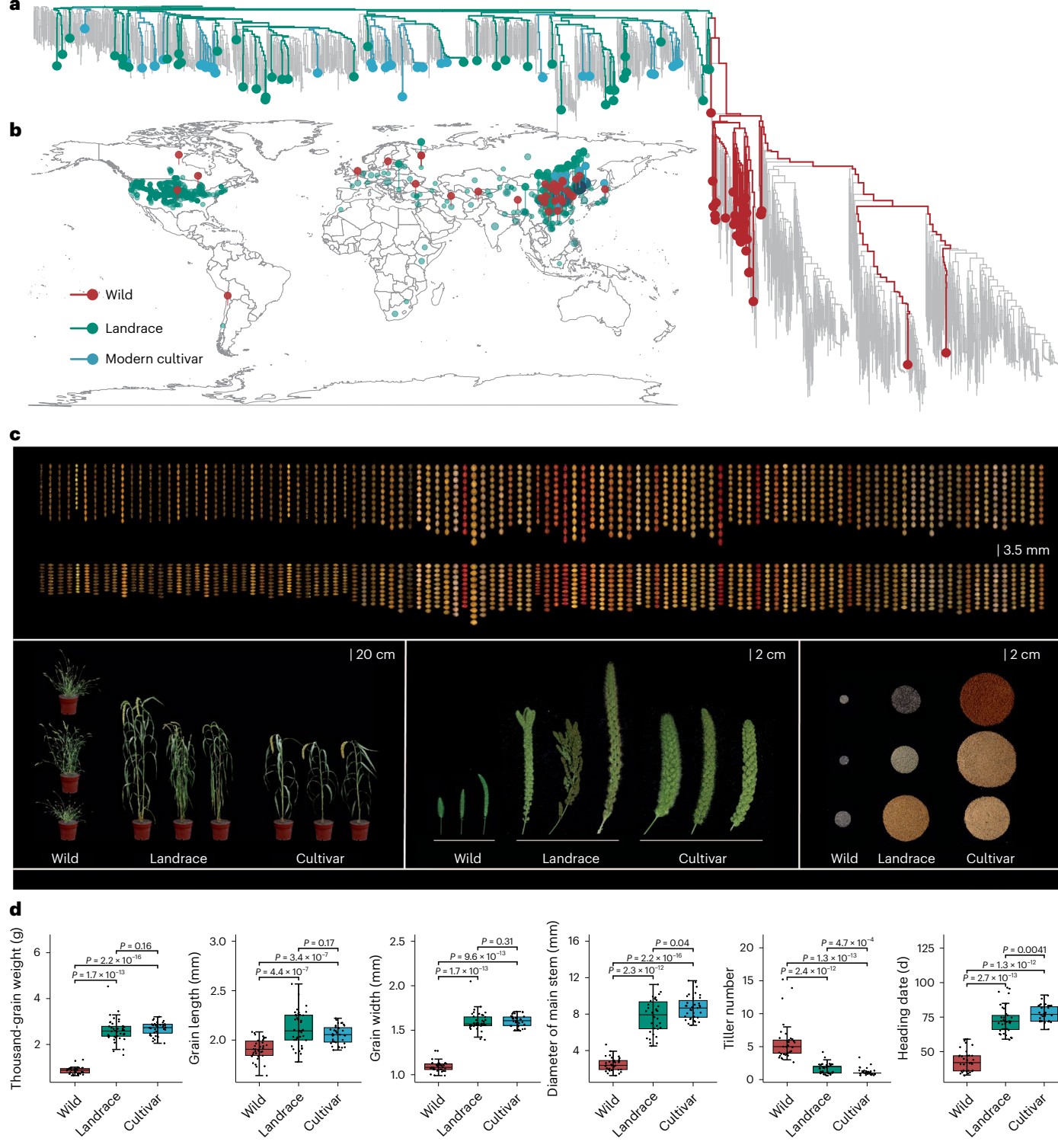

**Fig. 2 | The distribution and diverse phenotypes of 110 representative *Setaria* accessions. a**, Phylogenetic tree of the 1,844 *Setaria* accessions. Lines with different colors indicate the 110 accessions for de novo assembly as follows: wild (red), landrace (green) and cultivar (blue). **b**, Geographic distribution of the 110 diverse representative accessions among all 1,844 *Setaria* accessions. The color of points corresponds to **a**. The map was created using the map data function in ggplot2. **c**, GL and GW for 110 accessions, and characteristics of plant architecture, panicle shape/size and grain yield per panicle of representative wild, landrace and cultivar varieties of foxtail millet. **d**, Differences in TGW, GL, GW, diameter of main stem, tiller number and heading date for wild, landrace and modern cultivars. The number of samples in wild, landrace and cultivar in boxplots of **d** is 35, 40 and 35, respectively. In boxplots, the 25% and 75% quartiles are shown as lower and upper edges of boxes, respectively, and central lines denote the median. The whiskers extend to 1.5× the interquartile range. Significance levels are computed from two-sided Wilcoxon tests.

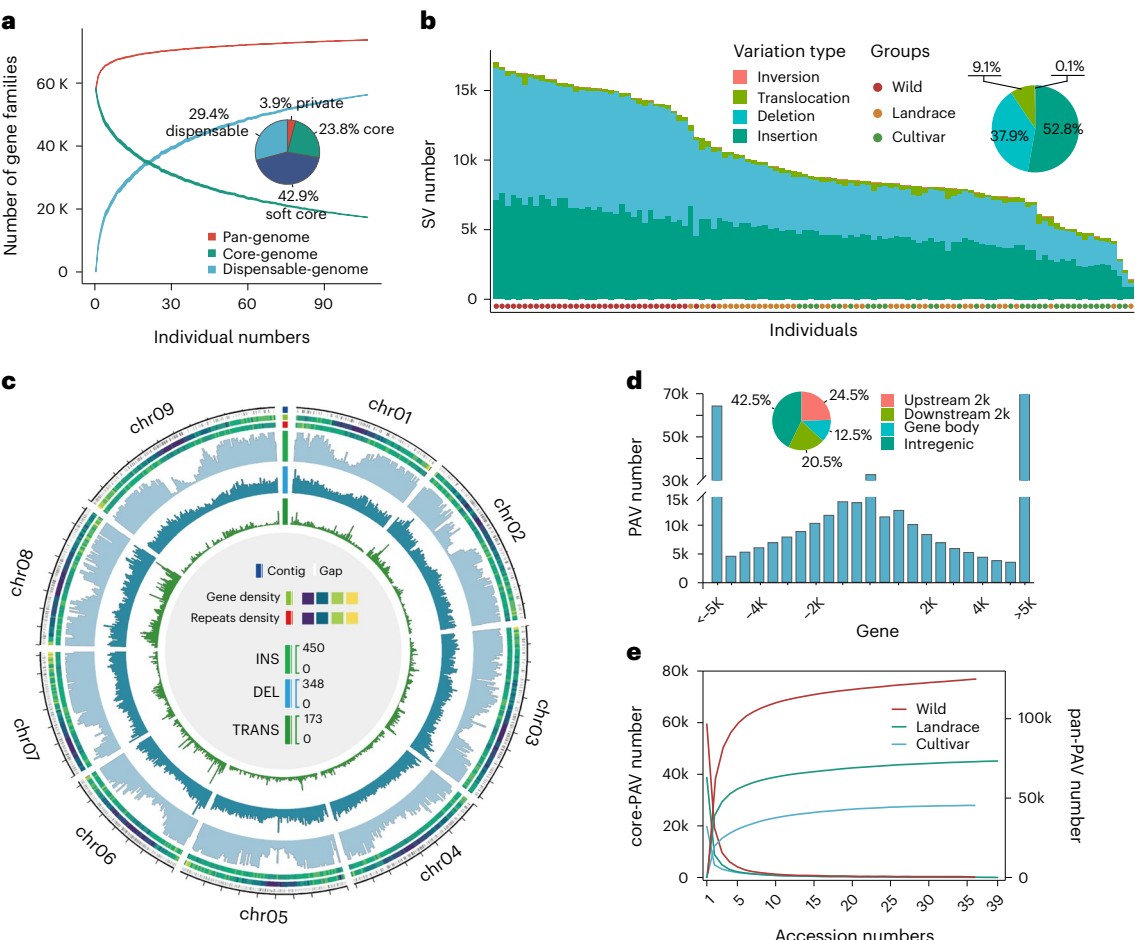

**Fig. 3 | Pan-genome and structure variation of *Setaria*. a**, The *Setaria* pan-genome. The detrended growth curve of the pan-genome indicates a closed pan-genome of *Setaria*. The pie chart shows the proportion of gene family marked by composition. **b**, Stacked bar graph of SV number and type from the 110 accessions. **c**, Distribution of SVs of 112 genomes across the nine chromosomes of foxtail millet. **d**, Distribution of PAV numbers against the distance to gene. **e**, Cumulative curves of pan-PAV and core-PAV in different groups with additional accessions added. The detrended growth curve of pan-PAV indicates a closed pan-PAV of *Setaria*.

It has long been noted that similar traits have evolved across distinct cereal crop species during domestication, and these domestication syndrome traits appear to be determined by similar genes in distinct cultivated lineages. Indeed, we find several domPAV genes that are associated with domestication in various cereal crop species, including the maize morphological domestication gene *tb1*, the rice flowering gene *Hd3*, the grain weight/shape genes *LG1* and *GW6a*, and the starch gelatinization temperature gene *SSII* (Supplementary Fig. 9). To further identify possible domestication-related loci, we screened for genome-wide selection signatures associated with foxtail millet domestication using SNP data with three different methods. From SNP-based selective sweep analysis, we find that genes responsible for agronomic traits such as homologs to *Hd1*, *TGW6* and eating/cooking quality gene *SBE2* were also under selection during domestication (Supplementary Fig. 10), consistent with foxtail millet possessing higher grain yield, better eating and cooking quality, and a longer growth period after its domestication from green foxtail. However, SNP-based methods recalled only 22.4% (328) of domPAV genes (Fig. 4e), suggesting that using PAV frequencies could be a complementary approach to SNP-based methods in identifying genes under positive selection. Together, these analyses identified pan-genome variation (that is, the presence or absence of genes/sequences) that may have important roles during foxtail millet domestication and improvement.

## PAV genes in domestication of nonshattering and grain yield

To further explore the role of PAVs in foxtail millet evolution, we looked closely at the following two key domestication traits in cereal crops: seed nonshattering and increased grain yield. Seed nonshattering is considered a key phenotype of domesticated cereal crops and is indeed used by archeologists as a critical marker of crop domestication[26,27]. To identify seed-shattering loci, we performed QTL analysis and bulked segregant analysis sequencing (BSA-seq) using an RIL population (Supplementary Notes 1–5), and three major QTLs (*qSH5.1*, *qSH5.2* and *qSH9.1*) controlling seed shattering in *Setaria* were identified (Supplementary Fig. 11b,c).

For *qSH5.1*, we find that the recently reported *Setaria* shattering-related gene *SvLes1* contains a 6.7-kb domPAV and is a candidate gene[16]. Using near-isogenic lines (NILs), we also fine-mapped and narrowed *qSH9.1* to an 87.3-kb region between markers M2 and M3, which contained *Seita.9G154300* (*sh1*, a homolog of the rice-shattering gene *OsSh1*; Supplementary Notes 1–5). Two NILs, NIL-*SH1* and NIL-*sh1^insert*, with similar plant architecture but a distinct shattering phenotype, further confirmed *sh1* as the *qSH9.1* locus in foxtail millet (Fig. 4g and Supplementary Fig. 12). The gene function of *sh1* was also independently proved in a transgenic study in ref. 28.

Haplotype analysis of both *sh1* and *SvLes1* supports previous studies that the insertions in *SvLes1* are not always involved in foxtail millet domestication[29], while the insertion in *sh1* is fixed in domesticated

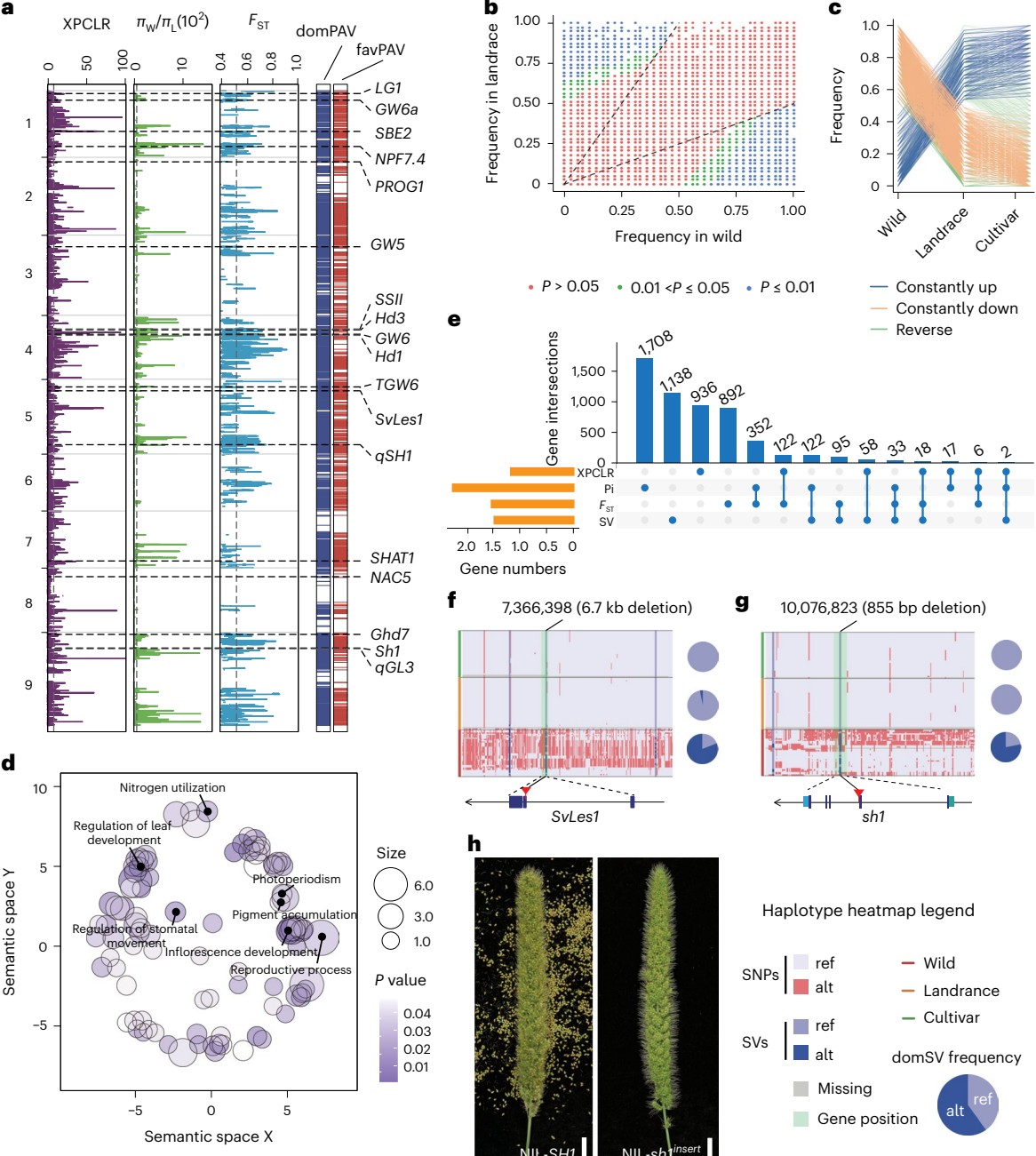

**Fig. 4 | GS signatures of foxtail millet domestication. a**, XPCLR, nucleotide diversity ratio ($\pi_W/\pi_L$), and $F_{ST}$ tests are used for selection analysis in *S. viridis*. Vertical dashed lines indicate genome-wide threshold of selection signals (XPCLR > 9.66, $\pi_W/\pi_L$ > 72.96 and $F_{ST}$ > 0.53). DomPAV and favPAV correspond to **b** and **c**. **b**, Scatter plots show PAV frequencies in landrace and wild (*P* value computed using two-sided Fisher's exact test). **c**, Frequency pattern of domestication-related PAVs (domPAVs). Lines in orange and blue indicate favPAVs during domestication. **d**, GO enrichment analysis of favPAV-genes. Color intensity (*P* value) reflects the significance of enrichment test (computed using two-sided Fisher's exact test). Circle size represents the frequencies of aggregated GO terms. **e**, Intersection of domestication-related genes across PAV-based and three SNP-based methods. **f**, Haplotype and selective signature at *SvLes1* gene. **g**, Haplotype and selective signature of *sh1* gene. **h**, Shattering phenotype of NIL with *SH1* and *sh1^insert* allele. Scale bar, 1.5 cm. $\pi_W/\pi_L$, $\pi_{wild}/\pi_{landrace}$.

foxtail millet (Fig. 4f,g). Interestingly, we found that neither the 6.7-kb deletion in *SvLes1* nor the 855-bp deletion in *sh1* was fixed in green foxtail (Fig. 4f,g), which suggests the action of other genes (for example, the gene located in *qSH5.2*) involved in the regulation of green foxtail shattering.

The second key domestication trait is increased grain yield in cultivated crop species[26,27] (Fig. 2c,d). Grain shape (grain width (GW) and grain length (GL)) is a key determinant of grain yield of foxtail millet, and correlation analysis and phenotypic distributions also suggest that

grain yield (thousand-grain weight (TGW)) is also determined by GW (Fig. 5a,b). To examine this trait genetically, we used the 110 high-quality genome sequences we developed, which are important resources for genome-wide association studies (GWAS) of domestication-related traits, encompassing accessions of both wild and cultivated forms. We performed an SV-based GWAS (SV-GWAS) for TGW, GW and GL. We find several significant GWAS signals on chromosomes 1, 3, 4, 5 and 9 for TGW and GW (Fig. 5c,d). Interestingly, we found a 366-bp deletion on chromosome 3, with the most significant association

with TGW ($P = 8.6 \times 10^{-15}$), and the second most significant association ($P = 7.3 \times 10^{-9}$) with GW (Fig. 5c,d). We also observed a moderate decline in nucleotide diversity in landraces in this region, and this deletion was classified as favPAV, suggesting positive selection during foxtail millet evolution (Figs. 4a and 5e).

We screened gene expression patterns in ten tissues from 'A10' (wild) and 'Yugu1' (cultivar). The 200-kb interval around this SV harbored 27 genes, eight of which showed differential expression patterns in seeds at the grain-filling stage between 'A10' and 'Yugu1' (Fig. 5f). We then searched for rice orthologs of these eight genes and found that *Seita.3G109700* was most likely to be the causal gene (hereafter, we named *SiGW3*) for TGW and GW; this locus has 73% sequence similarity with the rice domestication-regulation-related *GW5/GSE5* gene, which regulates rice grain size by influencing cell proliferation in spikelet hulls[30,31].

To validate *SiGW3* function, we overexpressed this gene in foxtail millet (accession 'Ci846'). Compared to wild-type plants, transgenic plants showed higher *SiGW3* gene expression, reduced TGW and GW and increased GL (Fig. 5g–k). To identify the causal variant, we analyzed genomic variants within *SiGW3* and a 20-kb region flanking the locus in the 110-millet accessions and found that only the 366-bp deletion (−7.2 kb away from the gene) cosegregated with the phenotype (Fig. 5l). Transient assays in foxtail millet protoplasts indicate that constructs with green foxtail distal sequences (wild-type) and modified foxtail millet distal sequence components excluding the 366-bp fragment (△C) drove higher luciferase reporter gene expression compared to constructs containing the 366-bp foxtail millet cultivar (C) fragment (Fig. 5m). This indicates that *SiGW3* negatively regulates grain weight, and the distal 366-bp genomic sequence possibly represses the expression of *SiGW3*, thereby increasing grain weight in domesticated foxtail millet. *SiGW3* has a similar function and selection pattern in both foxtail millet and rice[30] and also appears to be under strong selection in broomcorn millet (*Panicum miliaceum*; Fig. 5n), suggesting that the same gene may be involved in GW evolution in three different cereal grass lineages.

## Graph-based genome facilitates breeding of foxtail millet

To account for pan-genome variation and develop a key resource for breeding, we constructed a graph-based reference genome of *Setaria* by integrating 107,151 insertions, 76,915 deletions and 363 inversions across 112 foxtail millet and green foxtail accessions into the Yugu1 reference genome sequence (Methods). The availability of a graph-based genome sequence that goes beyond classical single-genome reference assemblies could capture more missing heritability.

We genotyped 1,844 *Setaria* accessions using Illumina short-read sequences and the graph-based genome and also collected 226 sets of phenotypes (68 traits) including yield, plant architecture, growth time, biomass, grain quality, coloration and disease resistance-related traits. To identify genes that operate across a broad set of climatic environments, we studied these traits at 13 distinct locations from 18.3°N (Sanya) to 47.3°N (Qiqihar) and 87.7°E (Urumqi) to 123.9°E (Qiqihar)

across 11 years (Fig. 6a, Supplementary Fig. 13 and Supplementary Table 12).

We find that most phenotypes were largely influenced by their field growing environments (Fig. 6b and Supplementary Table 13). To optimize breeding potential in different environmental conditions and more efficiently exploit genetic resources, we performed GWAS and genomic selection (GS) studies for all 226 phenotypes. We found that SV-based GWAS improves SNP-based GWAS efficiency for some traits (Fig. 6c,d). A total of 1,084 signals were identified to be substantially associated with 128 phenotypes for 60 traits, and 60 of the signals/QTL (5.5%) were only detected by SV-GWAS (Fig. 6d and Supplementary Table 14). Furthermore, linkage disequilibrium analysis showed that ~36.9% of SVs were not in LD with flanking SNPs (±50 kb, $R^2 < 0.5$) (Fig. 6e), which indicates that abundant genetic information associated with SVs are not captured by SNP markers.

We illustrate the utility of using graph-based genomes and associated SVs in GWAS mapping by examining a few traits. Apparent amylose content (AAC) is a key factor that affects eating and cooking quality in different crops, as determined by the granule-bound starch synthase gene (*GBSS/Waxy*)[32,33]. We directly identified the AAC-associated lead SV (a 196-bp insertion at position 1,485,625 on chromosome 4, $P < 1.39 \times 10^{-16}$) located 1.6 kb downstream from the *Seita.4G022400* (*GBSSI*) gene, while the lead SNP ($P < 5.64 \times 10^{-9}$) is found to be 398 kb away from the *GBSSI* gene (Supplementary Fig. 14).

We also found that two lead SVs, a 277-bp deletion in chromosome 1 and a 3.9-kb deletion in chromosome 2, were substantially associated with TGW ($P < 2.73 \times 10^{-6}$, Dingxi 2018) and peduncle length ($P < 4.67 \times 10^{-7}$, Changzhi 2011) through SV-GWAS, while no associated SNPs could be detected within a 50-kb interval of these SVs (Supplementary Figs. 15 and 16). Interestingly, we found a pleiotropic gene (*Seita.9G020100*), encoding a homolog of rice *Ghd7*, which has crucial roles in rice production and adaptation[34], and was only detected by SV-GWAS. Lead SVs are also substantially associated with heading date ($P < 5.99 \times 10^{-11}$, Beijing 2016), leaf length ($P < 3.92 \times 10^{-9}$, Anyang 2011), primary branch number ($P < 5.74 \times 10^{-10}$, Changzhi 2011) and straw weight ($P < 1.31 \times 10^{-6}$, Qitai 2014; Supplementary Fig. 17). Together, these indicate that SVs in foxtail millet may contain additional genetic information that are not represented by SNPs. It should be noted that some of these GWAS loci may have been under positive selection; of the 52 genomic regions associated with selection in cultivated subpopulations C1–C3 (Supplementary Table 4), eight regions overlap with GWAS hits for panicle number, branch number, emergence date, bristle color and grain glycine and arginine contents. We also find that for key domestication traits such as TGW and GW, all the GWAS signals span domPAVs, again linking these SVs to foxtail millet evolution.

Finally, we developed and evaluated the prediction accuracy of different marker panels for GS studies of the 68 agronomic and quality traits under geographically-distinct environments. With hundreds of SNPs and SVs, different phenotypes showed a range of predicted GS precision, with 97% of phenotypes with predicted precision over

**Fig. 5 | The *SiGW3* gene regulates grain yield of foxtail millet during domestication and improvement. a**, Phenotypic correlation between TGW, GL and GW. **b**, Phenotypic distribution of TGW, GL and GW. **c,d**, Manhattan plots of SV-GWAS for TGW and GW, respectively. The horizontal lines indicate Bonferroni-corrected genome-wide significance threshold ($\alpha = 1$ and $\alpha = 0.05$). **e**, Distribution of nucleotide diversity of wild, landrace and cultivar varieties in a 200-kb interval. **f**, Expression patterns of 27 genes within the 200-kb interval harboring the peak SV. **g**, The grain size difference of wild-type and *SiGW3* overexpression lines. **h–k**, Comparison of expression levels and TGW, GW and GL between wild-type Ci846 and three independent overexpression lines. **l**, Haplotype analysis of *SiGW3* and 20-kb left- and right-flanking genomic regions. The black arrows indicate three landraces with the same genotype as wild accessions at scaffold_3:7310555. **m**, Validation of function of the 366-bp deletion upstream of *SiGW3*. Transient assays are performed in foxtail millet leaf

protoplast. The construct backbone consists of the minimal promoter from the cauliflower mosaic virus (mpCaMV, green box), luciferase ORF (white box) and the nopaline synthase terminator (purple box). Portions of distal components of the control region (orange boxes) from foxtail millet cultivar and green foxtail (wild type) were cloned into restriction sites upstream of the minimal promoter. 'Δ' denotes excision of a 366-bp SV from the distal component. Horizontal blue bars show expression levels for each construct. The number of samples is 5. **n**, XPCLR, $F_{ST}$ and $\pi$ values between wild and cultivated broomcorn millet. Red-dashed lines are selection signals ($XPCLR > 53.6$, $F_{ST} > 0.644$). The vertical dashed line indicates the homologous gene *longmi029371* of *SiGW3* in broomcorn millet. Data are presented as mean ± s.d. in **h–k** and **m**; significance is computed by two-tailed Student's *t*-test. The number of samples in **h** and **i** is 6 and 3, respectively. The number of samples/seeds of WT, OE1, OE2 and OE3 in **j** and **k** is all 35.

0.7, and the highest prediction precision at more than 0.95 (leaf color of seedling in Beijing; Supplementary Table 15). We found that two traits have higher precision with SV-only markers compared to other marker subsets, and the precision of 167 (73.9%) traits with both SNP

and SV markers increased between 0.04% and 12.67% compared to SNP-only markers (Fig. 6f and Supplementary Table 15). To explore the breeding potential in foxtail millet, we estimated genomic estimated breeding values (GEBVs) using 1.04 million haplotype combinations

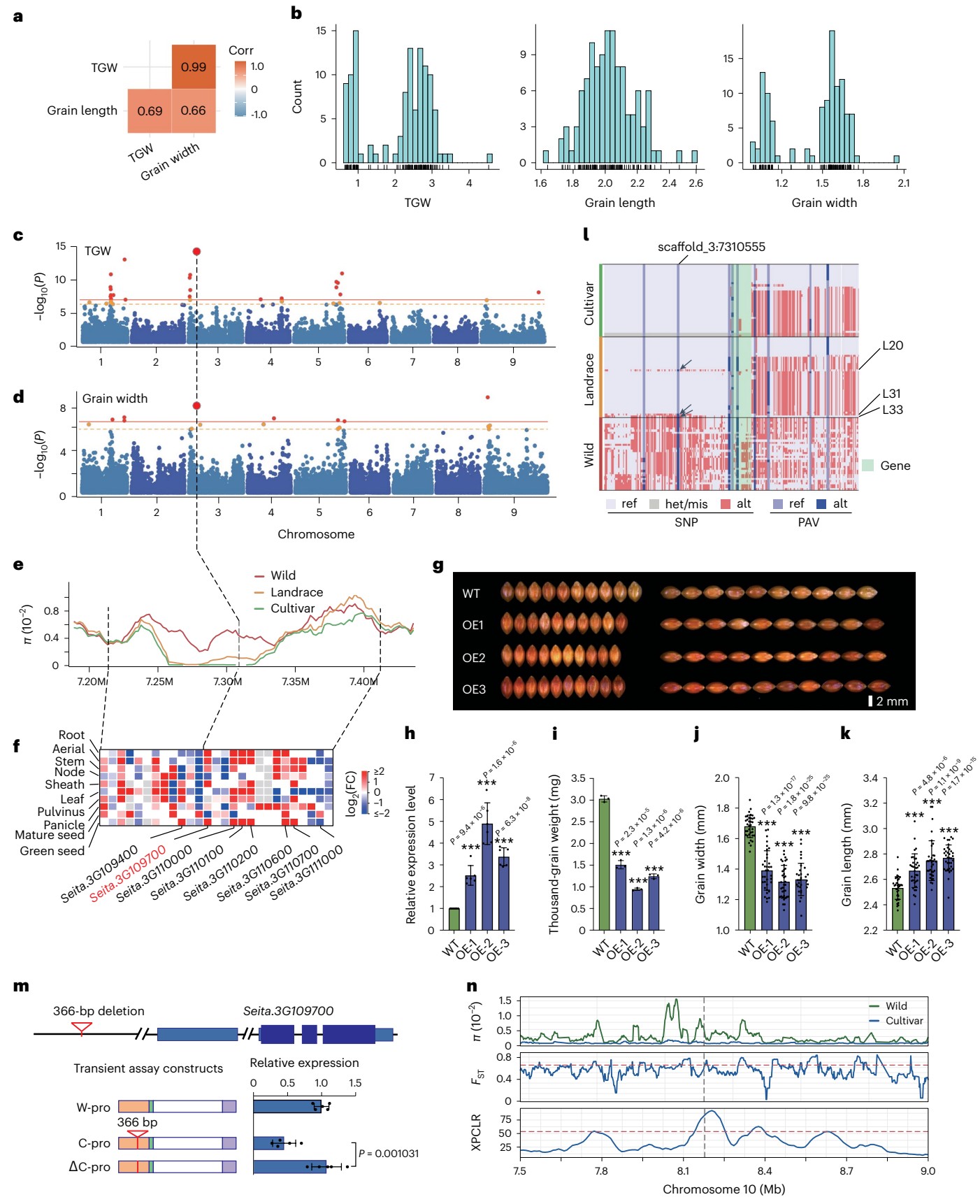

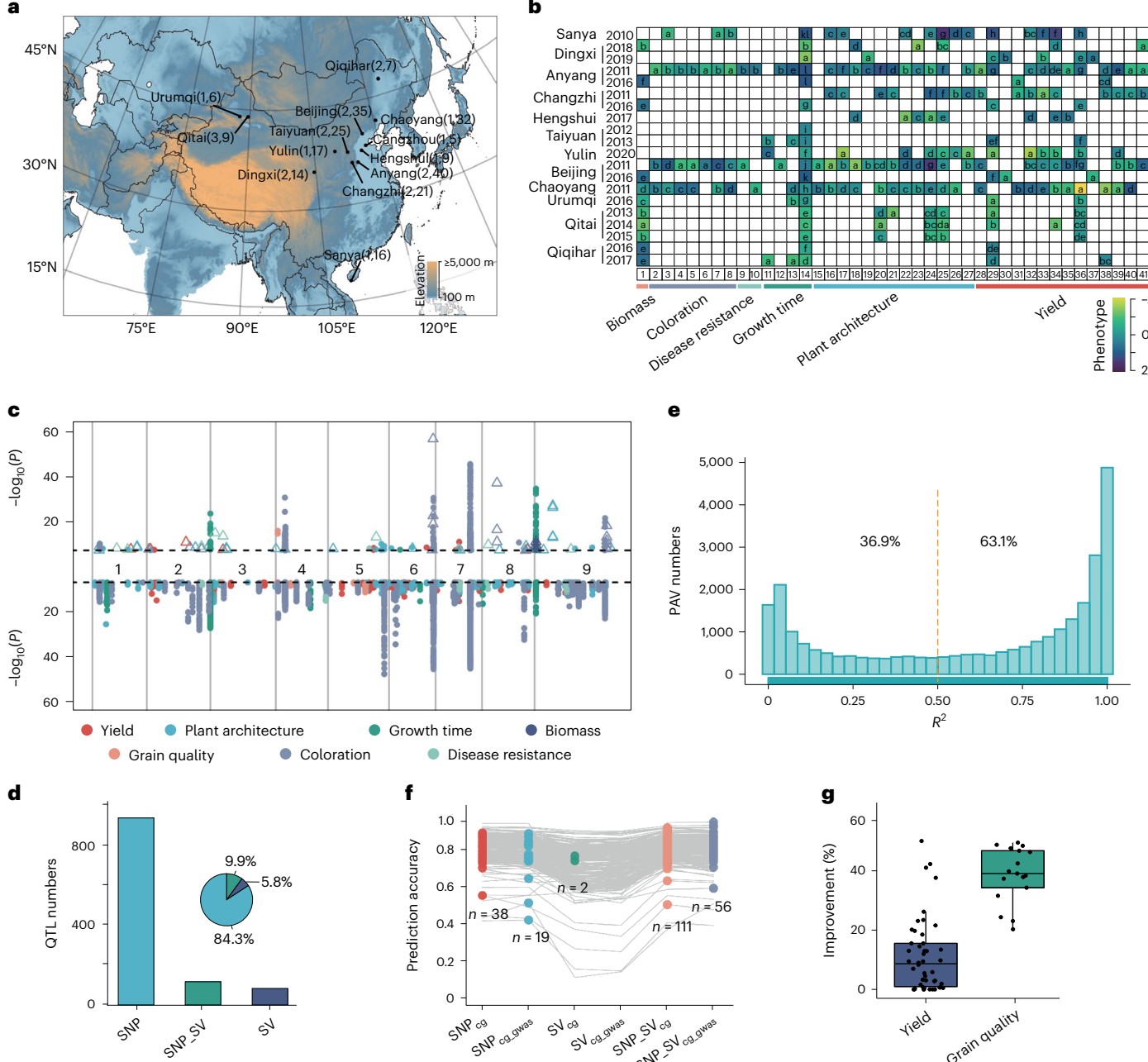

**Fig. 6 | Large-scale GWAS and genomic prediction for 247 sets of phenotypes using SV and SNP markers. a**, Phenotype collection from 13 geographic locations across 11 years. The numbers in parentheses are number of years and traits evaluated at corresponding locations. The map was created by the QGIS software with source data from the National Earth System Science Data Center, National Science & Technology Infrastructure of China. **b**, Phenotypic variation among different growth conditions. Different letters in heatmap represent significant differences ($P < 0.05$) according to Duncan's multiple comparisons test, which was conducted using two-sided ANOVA. Heatmap color represents the scaled phenotype values. Phenotypes from 1 to 41 correspond to Supplementary Table 13. **c**, Manhattan plots of SV-GWAS (top) and SNP-GWAS (bottom) of 247 sets of phenotypes. The dashed vertical lines indicate Bonferroni-corrected significance threshold ($\alpha = 0.05$). The triangles indicate

the associated signals only detected by SV-GWAS. **d**, Frequency of phenotype-associated loci detected by different markers. **e**, Linkage analysis between SVs from the graph-based genome using 680 accessions, and their nearby flanking ( ± 50 kb) SNPs. **f**, Precision of different phenotypes with different subsets of markers. Gray lines represent different phenotypes, and colored points denote the prediction precision with corresponding markers higher than others. Suffixes cg and gwas represent high-effect marker panels selected based on the feature importance by CropGBM and GWAS, respectively (Methods). **g**, Improvement percentage of yield ($n = 46$) and grain quality-related traits ($n = 17$) using base substitution of the top 20 highest effective variants. In boxplots, the 25% and 75% quartiles are shown as lower and upper edges of boxes, respectively, and central lines denote the median. The whiskers extend to 1.5× the interquartile range.

for phenotypes of 46 yield-related traits and 17 grain quality traits. Our results indicate that GEBVs of yield and grain quality traits could be improved by up to 50% and 49%, respectively (Fig. 6g and Supplementary Table 16).

## Discussion

Foxtail millet has been widely considered one of the founder crops in East Asia[1], whose wide environmental growing niche, C4 photosynthetic system, relatively small genome, short growing period and ease

of transformation make it a key crop species to deal with global food security amid changing world climates. The 110 core-set reference-level genomes we assembled represent the broad range of diversity in 1,844 *S. italica* and *S. viridis* accessions and ecotypes, and will serve as a critical resource for future biological studies and breeding efforts. With these genomes, we were able to establish a complete pan-genome and graph-based genome of *Setaria*, which offers insights into genomic variation across wild and cultivated *Setaria*, and provides valuable tools for functional genomic analyses and precision breeding in foxtail millet.

Our demographic analysis provides clues to the evolution of this important crop species. Our analysis identified the immediate ancestral progenitor subpopulation in green millet (W1), and based on the amount of drift (Supplementary Fig. 3a), suggested that C3, which can tolerate a wider range of climatic/environmental conditions, may have been established as the first of cultivated foxtail millet subpopulation. Enabled by the 110 de novo assembled *Setaria* genomes, we identified genomic regions that may be associated with foxtail millet domestication and improvement, providing genetic insights into how this domesticated species evolved.

SV identification has long been challenging when using short-read resequencing data. Nevertheless, the critical role of SVs in crop domestication, trait determination and agronomic improvement has been demonstrated in various studies[6–14]. With our constructed pan-genome comprising over 100 reference-level genome sequences, we identified ~10,000 SVs per *Setaria* genome, comparable with that seen in tomato[35] but fewer than in rice[8]. A substantial number of these SVs, particularly PAVs, were associated with TEs, consistent with TE activity being an important mechanism for SV generation in genomes[36,37]. The effect of PAVs in the genome also may differ across genes, and we find that indeed SVs are substantially found in lowly expressed genes. This pattern is also observed in rice[7,8] and is consistent with a stabilizing model of gene expression evolution[38], in which lowly expressed genes would be expected to be under weaker selection and thus more likely to be associated with PAVs[39,40]. Finally, similar to the studies of other crops, we find that SVs also underlie foxtail millet trait determination, exemplified by our study of two key domestication genes, *SiGW3* and *sh1*.

Construction of the graph-based genome allowed us to genotype SVs in a large population using short-read resequencing and to perform GWAS and GS in 680 foxtail millet accessions for 68 traits across 13 different geographic locations, each with distinct climatic growing conditions. We identified SNPs and SVs substantially associated with various phenotypes, which could be used in genomic prediction for foxtail millet in different environments. Indeed, the prediction precision for the majority of traits increased if both SNP and SV markers were jointly used, and we find two traits have higher precision with SV-only markers compared to SNP-only markers. This prediction accuracy is substantially higher than observed in tomato[12] possibly due to species or trait specificity. With our graph-based genome, we can also estimate potential breeding values of yield and grain quality-related traits, providing avenues for foxtail millet breeding for climate change adaptation.

Together, our investigation highlights the utility of analyzing crop pan-genomes to provide more complete catalogs of genetic variation, and together with the growing number of examples of SVs with genetic effects in other crops[6–14], we provide further evidence of the crucial role that pan-genome variants have in crop evolution and breeding. This may prove crucial in developing appropriate breeding programs for other crops, and help guide and accelerate crop improvement by marker-assisted breeding, GS and/or genome editing.

## Online content

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

[1]Institute of Crop Sciences, Chinese Academy of Agricultural Sciences, Beijing, China. [2]State Key Laboratory of Integrated Management of Pest Insects and Rodents, Institute of Zoology, Chinese Academy of Sciences, Beijing, China. [3]Center for Genomics and Systems Biology, New York University, New York City, NY, USA. [4]Shenzhen Branch, Guangdong Laboratory of Lingnan Modern Agriculture, Genome Analysis Laboratory of the Ministry of Agriculture and Rural Affairs, Agricultural Genomics Institute at Shenzhen, Chinese Academy of Agricultural Sciences, Shenzhen, China. [5]College of Agronomy, Northwest A & F University, Yangling, China. [6]College of Life Sciences, Shanxi Agricultural University, Taigu, China. [7]State Key Laboratory of Plant Physiology and Biochemistry & National Maize Improvement Center, Department of Plant Genetics and Breeding, China Agricultural University, Beijing, China. [8]School of Life Sciences, Institute of Life Sciences and Green Development, Hebei University, Baoding, China. [9]Anyang Academy of Agriculture Sciences, Anyang, China. [10]Center for Agricultural Genetic Resources Research, Shanxi Agricultural University, Taiyuan, China. [11]Research Institute of Cereal Crops, Xinjiang Academy of Agricultural Sciences, Urumqi, China. [12]Institute of High Latitude Crops, Shanxi Agricultural University, Datong, China. [13]Institute of Dry-Land Farming, Hebei Academy of Agricultural and Forestry Sciences, Hengshui, China. [14]Millet Research Institute, Shanxi Agricultural University, Changzhi, China. [15]Qiqihar Sub-Academy of Heilongjiang Academy of Agricultural Sciences, Qiqihar, China. [16]Cangzhou Academy of Agriculture and Forestry Sciences, Cangzhou, China. [17]Dingxi Academy of Agricultural Sciences, Dingxi, China. [18]Beijing Key Laboratory of Agricultural Genetic Resources and Biotechnology, Beijing Academy of Agriculture and Forestry Sciences, Beijing, China. [19]Center for Genomics and Systems Biology, New York University Abu Dhabi, Abu Dhabi, United Arab Emirates. [20]These authors contributed equally: Qiang He, Sha Tang, Hui Zhi, Jinfeng Chen. ✉e-mail: jiaguanqing@caas.cn; mp132@nyu.edu; diaoxianmin@caas.cn

## Methods

### Plant material and sequencing

All sequenced 1,004 foxtail millet and green foxtail accessions were purified for at least four generations in Beijing and Hainan, China. For sampling, we planted all accessions at the Experimental Station of the Institute of Crops Sciences, Chinese Academy of Agriculture Sciences, Beijing, in the 2018 growing season. For GWAS and GS analyses, we planted and examined agronomic and grain quality traits in 13 distinct environments at different years (listed in Supplementary Table 12).

Young leaves were collected and genomic DNA was extracted using cetyltrimethylammonium bromide (CTAB) and used to construct sequencing libraries following the manufacturer's instructions (Illumina Inc.). Libraries were paired-end (NGS) sequenced on Illumina NovaSeq 6000 at Novogene. For three representative accessions, long-read library construction followed standard protocol (Pacbio Inc.) and was sequenced on the Pacbio RSII platform at Nextomics Bioscience. Long-read library construction and sequencing for the other 107 de novo assembled accessions were performed by Berry Genomics with the Pacbio Sequel II platform (Supplementary Table 5).

Total messenger RNAs were extracted using TRIzol (Invitrogen) from different tissues and sequenced by the NovaSeq 6000 platform. For BioNano, fresh leaf tissues from 10-d-old seedlings of three accessions (Me34V, Ci846 and Yugu18) were collected and high-molecular-weight DNA was extracted and labeled according to standard protocols from BioNano Genomics. All labeled samples were loaded and analyzed using the BioNano Genomics SAPHYR system.

### SNP and SV calling of 1,844 accessions

Low-quality sequencing reads of the 1,844 accessions were removed using fastp (v0.23.0)[41] with default parameters, and filtered reads were mapped to the Yugu1 reference genome with BWA (v0.7.12-r1039)[42] using default parameters. Nonunique mapped and duplicated reads were excluded using SAMtools (v1.7)[43] and Genome Analysis Toolkit (GATK v4.1.4)[44], respectively. SNP calling was performed by GATK (v4.1.4)[44]. SnpEff (v5.0)[45] was used for annotating and predicting the effects of identified SNPs and indels. To identify structural variation in the 1,844 accessions, we mapped filtered Illumina short reads to the *Setaria* graph-based reference genome and genotyped SVs using vg toolkit (v1.28.0)[46] with default parameters.

### Phylogenetic and population structure analysis

Biallelic SNPs or PAVs with missing frequency <10% and minor allele frequency >0.05 were kept for phylogenetic analysis. SNP-based neighbor-joining phylogenetic tree was inferred using MEGA-CC (v10.1.8)[47] and SNPhylo (v2018-09-01)[48] with standard settings and 1,000 bootstrap values. SV-based maximum-likelihood phylogenetic tree was constructed based on binary PAV data with 1,000 bootstraps using IQ-TREE (v2.1.2)[49]. Phylogenetic trees were drawn using ggtree[50], an R package. We performed a population structure analysis using the ADMIXTURE (v1.3.0)[51] software, initially with $k$ ranging from 2 to 20. Here $k = 7$ was subsequently chosen because it was the minimal value of $k$ that separated all previously known groups of green foxtail[16]. We then ran ADMIXTURE ten times with varying random seeds at $k = 7$.

### Demographic history inference

Scripts for our population genomic analyses are deposited at https://github.com/qiangh06/Setaria-pan-genome/tree/main/Population%20genomic%20and%20Demographic%20inference. For demographic history analysis, we aimed at estimating the formation process of three subgroups of foxtail millet. For these analyses, we filtered SNPs with heterozygosity >0.05, minimum allele frequency <0.05 and genotyping rate <90% using PLINK (v.1.90)[52]. To reconstruct the evolutionary relationships between domesticated subpopulations C1–C3 and the closest wild population W1, we used Admixtools (v2.0)[23] on R v4.13 to construct an admixture graph with no migration edges. We used

a maximum absolute f4-statistic $z$-score (|$z$-score|) threshold of <3.0 for accepting models and added the remaining wild subpopulations W2–W4 sequentially to explore whether they could be incorporated with no migration edges. Population admixture graphs including all seven subpopulations were also inferred using TreeMix (v1.13)[22], with W3 as an outgroup. We used the GRoSS method[53] to scan the genome for positive selection along each branch of our four-population admixture graph that comprised W1, C1, C2 and C3.

### Sequencing and assembly of the 110 *Setaria* accessions

We assembled 110 diverse *Setaria* accessions using two approaches. For three high-quality reference genomes (Me34V, Ci846 and Yugu18), we used Illumina NovaSeq 6000 and PacBio RSII platforms (Supplementary Table 5) for sequencing, complemented with BioNano optical maps. We estimated the genome size of these three accessions to be ~430 Mb according to the $k$-mer distribution of Illumina short reads. Over 50 Gb PacBio subreads (>100×; Supplementary Table 5) of each accession were subsequently assembled into contigs by CANU (v2.2)[24] and HERA (v1.0)[25]. After polishing with Illumina reads and further correction with BioNano physical maps, we obtained 75, 114 and 103 contigs for Me34V (398,819,634 bp, N50 = 21.1 Mb), Ci846 (412,045,876 bp, N50 = 21.0 Mb) and Yugu18 (409,028,184 bp, N50 = 20.6 Mb), respectively. For the other 107 accessions, we sequenced using Illumina NovaSeq 6000 at >40× short-read data (except Zhaogu1 with 37.5× data) for each accession. We examined genome size and heterozygosity using Jellyfish (v2.3.0)[54] and GenomeScope (v2.0)[55]. Based on examined genome heterozygosity, we generated >50× and >80× long-read data for low heterozygosity (<0.3%) and high heterozygosity (≥0.3%) accessions by the Pacbio Sequel II platforms, respectively (Supplementary Table 5). We subsequently de novo assembled these *Setaria* genomes using CANU[24] and HERA[25] pipelines. Self-alignment of whole-genome contig sequences was performed using default parameters of BWA-MEM (v0.7.12-r1039)[42], and heterozygous sequences were filtered with Redundans (with -t 10, -identity 0.55, -overlap 0.80, --noscaffolding, and -nogapclosing) and Purge Haplotigs (with default parameters). Overlaps between contig sequences were merged using the results of BWA-MEM self-alignment.

NGS data were mapped to the genome using BWA-MEM (v0.7.12-r1039)[42], and the results were filtered with Q30 by SAMtools (v1.7)[43]. Finally, the genome sequence was corrected using Pilon (v1.22)[56] based on filtered alignments. Three rounds of genome correction were performed by Pilon. Finally, contigs were aligned to the reference genome to construct pseudo-chromosomes using Mummer (v4.0)[57] with the parameters '-mum -mincluster = 1000'.

### Evaluation of genome assemblies

We assessed the completeness of the genic region of assemblies using BUSCO (v5.2.0)[58] with 1,440 embryophyte genes. To assess the assembly completeness of intergenic regions, we used the LAI using LTR_retriever (v2.9.0)[59]. We also assessed genome completeness by mapping high-quality Illumina short reads to the corresponding assembly using BWA (v0.7.12-r1039)[42] with default parameters. $K$-mer-based completeness, quality and false duplication evaluation were performed by Merqury (v1.3)[60].

### Repeat annotation

A combination of ab initio and homology-based methods was used to annotate repeats in the assembled genomes. First, we constructed an ab initio repeat library using LTR_FINDER (v1.05)[61] and RepeatModeler (v4.0.6)[62] with default parameters. The predicted repeat library was aligned with the PGSB repeater database[63] to assign repeats into distinct families. Next, Repbase (v20.11) was used to conduct homology-based annotation using RepeatMasker (v1.0.10)[64]. Finally, overlapping repeat sequences that belong to the same repeat class were combined. For overlapping repeats belonging to different repeat classes, overlapping

regions were divided. In addition, Tandem Repeats Finder[65] was used to annotate tandem repeats.

## Prediction and functional annotation of protein-coding genes

We used transcriptome data from whole plants of three representative accessions (wild, Me34V; landrace, Ci846; and modern cultivar, Yugu18). RNA-seq data from each accession were separately assembled using Trinity (v2.8.5)[66] with default parameters. Assembled transcripts of Me34V, Ci846 and Yugu18 were used for annotation of wild, landrace and modern cultivars, respectively. Each genome was annotated to obtain gene models using UniProt SwissProt (v2020_01)[67] protein database and MAKER (v3.01.03)[68]. These genes were used to train Augustus (v3.2.3)[69] and SNAP (v2006-07-28)[70], and the resulting training sets were used for annotation of corresponding genomes. Assembled transcripts were used as EST evidence, and protein sequences of rice (MSU v7)[71], *Arabidopsis thaliana* (TAIR10)[72], maize (B73 RefGen_v4)[73], sorghum (v3.1.1)[74], foxtail millet (v2.2)[5,18], green foxtail (v2.1)[16] and UniProt SwissProt database (release-2017_01) were used as protein evidence. Using models trained by SNAP and Augustus, the second round of gene annotation was performed for all repeat-masked genomes, and genes with AED < 0.4 were kept. Functional annotation of predicted genes was performed using InterProScan 5.0 (ref. [75]) to assign Gene Ontology (GO) and Kyoto Encyclopedia of Genes and Genomes (KEGG) terms. Based on the results of functional annotation, TE-related genes were filtered.

## Gene-based pan-genome construction

We aligned the CDS of all annotated genes to the 108 genomes from cultivated and wild (W1) foxtail millet using GMAP (v2015-09-21)[76]. If a gene was aligned with >99% coverage and identity, it was considered present in the corresponding genome. We performed a pan-genome analysis based on a Markov clustering approach[77]. All-versus-all comparisons were performed using diamond (v0.9.25)[78] with an *E*-value cutoff of $1 \times 10^{-5}$. Subsequently, all paired genes were clustered using OrthoFinder (v2.3.12)[77]. Based on their frequency, we classified genes into the following four categories: core (these present in all 111 individuals), soft core (these present in >90% of samples but not all; 100–110 individuals), dispensable (these present in more than one but less than 90%; 2–99 individuals) and private (present in only one accession).

## Identification of structural variation and graph-based genome construction of *Setaria*

We used the SyRI[79] pipeline for structural variation (insertion, deletion, translocation and inversion) identification in the 112 genomes. We first aligned each assembled genome to the Yugu1 reference genome using Minimap2 (v2.21-r1071)[80]. Raw alignment results were further used for variation calling using the SyRI (v1.2)[79] software with default parameters. We then filtered SVs with variant size of over 50 bp. From filtered results, insertions and deletions were treated as PAVs. We used the vg toolkit (v1.28.0)[46] for graph-based genome construction. First, we identified large PAVs and inversions with MUMmer (v4.0)[57]. Then, PAVs together with inversions detected by SyRI were integrated into the Yugu1 linear reference genome using the vg toolkit[46].

## Genomic selection signature identification

We used three different strategies, nucleotide diversity, $F_{ST}$ and XPCLR, for identifying selective sweeps based on high-quality SNP markers (MAF ≥ 0.05 and missing <0.1). For nucleotide diversity and $F_{ST}$ analysis, we used VCFtools (v0.1.17)[81] with 20-kb sliding and 2-kb step size. We performed XPCLR analysis using the XPCLR program (https://github.com/hardingnj/xpclr).

## Genome-wild association studies and identification of candidate genes in the GWAS-associated loci

We performed GWAS for 226 phenotypes in 680 accessions using high-quality SV and SNP markers (MAF ≥ 0.05 and missing <0.1) using

the Mixed-Model Association eXpedited program (EMMAx, v20120210) with the first ten PCAs as a random effect matrix. An effective number of independent makers (SNP and SVs) were estimated to be 640,288, and we defined the significance threshold by Bonferroni-corrected genome-wide significance ($\alpha = 0.01$).

For candidate gene identification, we used the following strategies: first, we grouped all associated SNPs/SVs ($P \leq 7.81 \times 10^{-8}$, Bonferroni-corrected genome-wide significance threshold ($\alpha = 0.01$)) of each phenotype into one cluster if the distance between the SNPs/SVs and the leading SNPs/SVs is ≤50 kb and the LD $R^2 \geq 0.3$. The grouped SNPs/SVs were defined as associated loci and represented by the leading SNPs/SVs. Second, we selected candidate genes in ±50 kb interval of leading SNPs/SV if their homologous gene was functionally related to corresponding phenotypes in rice or maize.

## High-effect marker panel selection and genomic prediction

First, we performed a feature selection analysis of three different marker panels (SNP panel, 2,711,024 SNPs; SV panel, 44,869 SVs; and SNPSV panel, 2,711,024 SNPs plus 44,869 SVs) for each of the 226 phenotype datasets independently using the CropGBM (v1.1.2)[82] software to estimate feature gain (FG)/marker effect of each SNP and SVs via information gain analysis. Second, highly effective markers were identified if their reduction of FG (ROF = $1 - FG_{max}/FG_i$, where $FG_{max}$ represents the highest FG value of the markers, and $FG_i$ represents the FG value of *i*th marker) was less than 0.99. Next, for each trait, we grouped markers into the following six panels: $SNP_{cg}$ panel contained highly effective SNP makers selected with ROF ≤ 0.99; $SNP_{cg\_gwas}$ panel was the union set of highly effective SNP makers selected with ROF ≤ 0.99 and significantly associated SNP markers from GWAS ($P \leq 7.81 \times 10^{-8}$); $SV_{cg}$ panel contained highly effective SV makers selected with ROF ≤ 0.99; $SV_{cg\_gwas}$ panel was the union set of highly effective SV makers selected with ROF ≤ 0.99 and substantially associated SV markers from GWAS ($P \leq 7.81 \times 10^{-8}$); $SNPSV_{cg}$ panel contained highly effective SNP and SV makers selected with ROF ≤ 0.99; and $SNPSV_{cg\_gwas}$ panel was the union set of highly effective SNP and SV makers selected with ROF ≤ 0.99 and substantially associated SV markers from GWAS ($P \leq 7.81 \times 10^{-8}$, Bonferroni-corrected genome-wide significance threshold ($\alpha = 0.01$)).

The predictive precision of models was assessed for each marker panel and corresponding phenotypes using Pearson's correlation between observed phenotypes and predicated GEBVs. We randomly divided the dataset into 580 and 100 lines for validation. The 580 lines were used as training sets to estimate marker effects, which were then used to predict GEBVs for the remaining 100 lines; this was replicated 100 times for each dataset.

## Breeding potential prediction

We used 63 datasets (7 yield and 17 grain quality-related traits in different environments) for breeding potential prediction. The marker panel with the highest prediction precision for the corresponding phenotype was selected. We then simulated 1.04 million haplotype combinations using the top 20 high-effective markers of accessions with the highest GEBVs. The improvement percentage of each phenotype was calculated by $\frac{GEBV\,max\_haplotype - GEBV\,max\_cultivated}{GEBV\,max\_cultivated} \times 100\%$, where $GEBV_{max\_haplotype}$ represents the highest GEBV of simulated haplotypes, and $GEBV_{max\_cultivated}$ denotes the highest GEBV of cultivated foxtail millet.

## Functional characterization of *SiGW3*

To generate overexpression constructs, a full-length coding sequence of *SiGW3* was amplified from green foxtail accession 'A10' and cloned into pCAMBIA1305 under the control of the ubiquitin (UBI) promoter. Primers OE-GW3-F and OE-GW3-R were used (Supplementary Table 17). SiGW3-OE vector was transformed into foxtail millet variety Ci846 by *Agrobacterium tumefaciens*-mediated transformation using strain EHA105. Three independent transgenic overexpression lines of *SiGW3*

**Article**

were identified and selfed to T3 generation. The expression of transgenic overexpression lines was further verified by qRT-PCR using primers listed in Supplementary Table 17. qRT-PCR experiment was conducted as described previously[20]. Around 200 seeds of WT and three independent transgenic lines were randomly selected, and photographed and measured by Wseen seed measurement instrument SC-G.

To validate the effect of 366-bp SV in the promoter of *SiGW3* on gene expression, we employed a dual-LUC transient expression assay using *Nicotiana benthamiana* leaves. Renilla luciferase (REN) reporter gene driven by the minimal 35S promoter was used as an internal control, and firefly luciferase (LUC) driven by the target 366-bp insertion promoter and the target 366-bp deletion promoter was amplified from *Setaria* wild species 'A10' and cultivar 'Yugu1', respectively. Primers used for amplifying the SV in *SiGW3* promoter sequences are listed in Supplementary Table 17. Three constructed vectors were then transformed into *Agrobacterium* GV3101 and co-infiltrated into leaves of 4-week-old *N. benthamiana*. Luciferase signals were imaged using Tanon 5200 and measured using Dual-Luciferase Reporter Assay System (E1910) kit (Promega) and Varioskan LUX (Thermo Fisher Scientific). Each measurement was conducted with five biological replicates. All reagents used in this study are listed in Supplementary Table 18.

### Geographic map generation
The geographical location information of the collection sites of all varieties and phenotypes in this study are marked on the map using ggplot2 (ref. 83) package in R (v4.1.0) and QGIS (v3.16)[84] software. The elevation map source data are collected from the National Earth System Science Data Center, National Science and Technology Infrastructure of China (http://www.geodata.cn/data/datadetails.html?dataguid=78789&docid=4850).

### Reporting summary
Further information on research design is available in the Nature Portfolio Reporting Summary linked to this article.

## Data availability
All long-read sequencing data and three Bionano cmap files have been deposited in the National Center for Biotechnology Information database under accession code BioProject PRJNA675302. All 110 assembled genomes and annotations were deposited at https://www.zenodo.org/record/7367881. 1,004 NGS resequencing data generated have been deposited in the NCBI database under accession code BioProject PRJNA841774 and PRJNA842100. Other 294 foxtail millet and 594 green foxtail whole-genome sequencing data were downloaded from NCBI (BioProject PRJNA636263, PRJNA560514 and PRJNA265547). The phenotypes used in GWAS and GS studies have been deposited in https://doi.org/10.5281/zenodo.7755340. Source data are provided with this paper.

## Code availability
All codes associated with this project are available at Github (https://github.com/qiangh06/Setaria-pan-genome) and Zenodo (https://doi.org/10.5281/zenodo.7743007)[85].

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

## Acknowledgements

The authors appreciate critical comments and advice from N. Stein (Leibniz Institute of Plant Genetics and Crop Plant Research) and J. Jia (CAAS). The authors thank H. Lu (State Key Laboratory of Rice Biology, China National Rice Research Institute, CAAS) and J. Gao (Hainan Academy of Ocean and Fisheries Sciences) for their helpful technical support on genome assembly and project discussion. The authors thank K. Xie (Guangzhou Genedenovo Biotechnology Co., Ltd.) for useful comments on demographic inference studies. We thank L. Yin (ICS Bioinformatics Group) for providing computing support. This work was supported by grants from the National Key Research and Development Program of China (2021YFF1000100), the National Key R&D Program of China (2019YFD1000700/2019YFD1000701 and 2018YFD1000700), the National Natural Science Foundation of China (31871692 and 31871630), the China Agricultural Research System (CARS-06-13.5), the Agricultural Science and Technology Innovation Program of Chinese Academy of Agricultural Sciences, Strategic Priority Research Program of Chinese Academy of Sciences (grant XDPB16), the US National Science Foundation Plant Genome Research Program (IOS-1546218 and 2204374) and the Zegar Family Foundation and the NYU Abu Dhabi Research Institute.

## Author contributions

X.D. conceived and designed the research. Q.H., S.T., H. Zhi., H. Liang., H.W. and G.J participated in material preparation. Q.H., H.D., J.S. and J.L. contributed to the genome assembly and annotation. Q.H. performed genomic variant calling, selective signature identification, genome-wide association study and genomic prediction. Q.H., X.L., J.Z., O.A. and M.P. performed population genetics analysis. Q.H. and J.Z. performed gene expression, functional enrichment and phenotypic data cleaning. S.T. contributed to QTL mapping of *sh1*. S.T., H. Zhang., L.X., W.Z. and H.W. contributed to the functional characterization of the *SiGW3*. S.T., H.Z., L.W., L.X., H.Y., Z.S., J.L., H.W., X.T., Z.Q., G.F., R.G., W.Z., Y.R., H.H., M.L., A.Z., E.G., F.Y., Q.L., Y.L., B.T., X.Z., R.J., B.F., J.Z. and J.W. planted the materials and collected phenotypic data at different geographic locations. Q.H., M.P. and X.D. oversaw the integration and conceptualization of results and wrote the manuscript. S.T., H. Li., P.Y., J.C. and G.J. revised the manuscript. All authors read, edited and approved the manuscript.

## Competing interests

The authors declare no competing interests.

## Additional information

**Correspondence and requests for materials** should be addressed to Guanqing Jia, Michael Purugganan or Xianmin Diao.

| | |
|---|---|

# Reporting Summary

## Statistics

For all statistical analyses, confirm that the following items are present in the figure legend, table legend, main text, or Methods section.

| n/a | Confirmed | |
|---|---|---|
| ☐ | ☒ | The exact sample size (*n*) for each experimental group/condition, given as a discrete number and unit of measurement |
| ☐ | ☒ | A statement on whether measurements were taken from distinct samples or whether the same sample was measured repeatedly |
| ☐ | ☒ | The statistical test(s) used AND whether they are one- or two-sided *Only common tests should be described solely by name; describe more complex techniques in the Methods section.* |
| ☒ | ☐ | A description of all covariates tested |
| ☐ | ☒ | A description of any assumptions or corrections, such as tests of normality and adjustment for multiple comparisons |
| ☐ | ☒ | A full description of the statistical parameters including central tendency (e.g. means) or other basic estimates (e.g. regression coefficient) AND variation (e.g. standard deviation) or associated estimates of uncertainty (e.g. confidence intervals) |
| ☐ | ☒ | For null hypothesis testing, the test statistic (e.g. $F$, $t$, $r$) with confidence intervals, effect sizes, degrees of freedom and $P$ value noted *Give P values as exact values whenever suitable.* |
| ☒ | ☐ | For Bayesian analysis, information on the choice of priors and Markov chain Monte Carlo settings |
| ☒ | ☐ | For hierarchical and complex designs, identification of the appropriate level for tests and full reporting of outcomes |
| ☒ | ☐ | Estimates of effect sizes (e.g. Cohen's *d*, Pearson's *r*), indicating how they were calculated |

*Our web collection on statistics for biologists contains articles on many of the points above.*

## Software and code

Policy information about availability of computer code

| Data collection | Data were sequenced from PacBio Sequel and Illumina NovaSeq 6000. Publicly available data were downloaded and used. |
|---|---|
| Data analysis | We used publicly available and appropriately cited software as described. No commercial software or code was used in this study. Software are listed as follows: fastp (v0.23.0), BWA (v0.7.12-r1039),SAMtools (v1.7),GATK (v4.1.4),SnpEff (v5.0),vg toolkit (v1.28.0),MEGA-CC (v10.1.8),SNPhylo (v2018-09-01),IQ-TREE (v2.1.2),ADMIXTURE (v1.3.0),PLINK (v.1.90),Admixtools (v2.0),TreeMix (v1.13),CANU (v2.2),HERA (v1.0),Pilon (version 1.22), IrysSolve (v3.5_12162019, https://bionanogenomics.com/support/software-downloads/),Mummer (v4.0),BUSCO (v5.2.0),LTR_retriever (v2.9.0),Merqury (v1.3),LTR_FINDER (v1.05),RepeatModeler (v4.0.6),RepeatMasker (v1.0.10),EDTA (v1.9.4),GMAP (v2015-09-21),diamond (v0.9.25),OrthoFinder (v2.3.12),SyRI(v1.2),minimap2 (v2.21-r1071),EMMAX (v20120210),R (v4.03),VCFtools (v 0.1.17),XP-CLR(v1.1.2,https://github.com/hardingnj/xpclr),CropGBM(v1.1.2). Custom codes are available at github: https://github.com/qiangh06/Setaria-pan-genome and Zenodo: https://doi.org/10.5281/zenodo.7743007. |

For manuscripts utilizing custom algorithms or software that are central to the research but not yet described in published literature, software must be made available to editors and reviewers. We strongly encourage code deposition in a community repository (e.g. GitHub). See the Nature Portfolio guidelines for submitting code & software for further information.

## Data

Policy information about <u>availability of data</u>

All manuscripts must include a <u>data availability statement</u>. This statement should provide the following information, where applicable:

- Accession codes, unique identifiers, or web links for publicly available datasets
- A description of any restrictions on data availability
- For clinical datasets or third party data, please ensure that the statement adheres to our <u>policy</u>

All long-read sequencing data and three Bionano cmap files have been deposited in National Center for Biotechnology Information database under accession code BioProject: PRJNA675302. All 110 assembled genomes and annotations were deposited at https://www.zenodo.org/record/7367881. 1004 NGS re-sequencing data generated have been deposited in the NCBI database under accession code BioProject: PRJNA841774 and PRJNA842100.  Other 294 foxtail millet and 594 green foxtail whole genome sequencing data were downloaded from NCBI (BioProject PRJNA636263, PRJNA560514 and PRJNA265547).The phenotypes used in GWAS and GS studies have been deposited in https://doi.org/10.5281/zenodo.7755340.

## Human research participants

Policy information about <u>studies involving human research participants and Sex and Gender in Research.</u>

| | |
|---|---|
| Reporting on sex and gender | *Use the terms sex (biological attribute) and gender (shaped by social and cultural circumstances) carefully in order to avoid confusing both terms. Indicate if findings apply to only one sex or gender; describe whether sex and gender were considered in study design whether sex and/or gender was determined based on self-reporting or assigned and methods used. Provide in the source data disaggregated sex and gender data where this information has been collected, and consent has been obtained for sharing of individual-level data; provide overall numbers in this Reporting Summary.  Please state if this information has not been collected. Report sex- and gender-based analyses where performed, justify reasons for lack of sex- and gender-based analysis.* |
| Population characteristics | *Describe the covariate-relevant population characteristics of the human research participants (e.g. age, genotypic information, past and current diagnosis and treatment categories). If you filled out the behavioural & social sciences study design questions and have nothing to add here, write "See above."* |
| Recruitment | *Describe how participants were recruited. Outline any potential self-selection bias or other biases that may be present and how these are likely to impact results.* |
| Ethics oversight | *Identify the organization(s) that approved the study protocol.* |

Note that full information on the approval of the study protocol must also be provided in the manuscript.

# Field-specific reporting

Please select the one below that is the best fit for your research. If you are not sure, read the appropriate sections before making your selection.

☒ Life sciences ☐ Behavioural & social sciences ☐ Ecological, evolutionary & environmental sciences

For a reference copy of the document with all sections, see nature.com/documents/nr-reporting-summary-flat.pdf

# Life sciences study design

All studies must disclose on these points even when the disclosure is negative.

| | |
|---|---|
| Sample size | We selected 110 representative Setaria accessions, including 35 wild, 40 landrace, and 35 modern cultivated accessions. The logic of this selection was based on phylogenetic relationships and geographic distribution, breeding and/or research contribution, and subgroups distributions to ensure that they are representative of genetic diversity within foxtail millet and green foxtail. |
| Data exclusions | No samples were excluded in this study. Filters applied to eliminate low-quality sequencing data and genetic variants were properly described in the Method section. |
| Replication | Five biological replicates were used in the qRT-PCR experiment. Three independent T3 transgenic lines were generated for the estimation of thousand grain weight, grain width, and grain length, in which three independent wild-type plants were also measured. All replications were successful and were used. |
| Randomization | For each foxtail millet or green foxtail individual, the sampling process for genome DNA/RNA sequencing was randomly conducted. All WT and transgenic plants were exposed to the same growth condition and treatment. For phenotype data collection at GWAS and GS studies, we randomly selected 3-5 plants for data collection per accessions in the filed. |
| Blinding | Blinding is not necessary for genome sequencing and assembly, since the investigators know which Setaria accessions they were handling. The investigators were blinded to group allocation during collecting data from WT and transgenic lines. |

# Reporting for specific materials, systems and methods

We require information from authors about some types of materials, experimental systems and methods used in many studies. Here, indicate whether each material, system or method listed is relevant to your study. If you are not sure if a list item applies to your research, read the appropriate section before selecting a response.

| Materials & experimental systems | | Methods | |
|---|---|---|---|
| **n/a** | **Involved in the study** | **n/a** | **Involved in the study** |
| ☒ ☐ | Antibodies | ☒ ☐ | ChIP-seq |
| ☒ ☐ | Eukaryotic cell lines | ☒ ☐ | Flow cytometry |
| ☒ ☐ | Palaeontology and archaeology | ☒ ☐ | MRI-based neuroimaging |
| ☒ ☐ | Animals and other organisms | | |
| ☒ ☐ | Clinical data | | |
| ☒ ☐ | Dual use research of concern | | |

