## [Peer Review File · Nature Genetics]

Peer Review Information

Manuscript Title: A graph-based genome and pan-genome variation of the model plant Setaria

Corresponding author name(s): Guanqing Jia, Professor Xianmin Diao, Professor Michael Purugganan

Reviewer Comments & Decisions:

Decision Letter, initial version:

28th Sep 2022

Dear Professor Purugganan,

Your Article, "A graph-based genome and pan-genome variation of the model plant Setaria" has now been seen by 3 referees. You will see from their comments below that while they find your work of interest, some important points are raised. We are interested in the possibility of publishing your study in Nature Genetics, but would like to consider your response to these concerns in the form of a revised manuscript before we make a final decision on publication.

To guide the scope of the revisions, the editors discuss the referee reports in detail within the team with a view to identifying key priorities that should be addressed in revision. In this case, we think all three referees have provided constructive reviews aimed at strengthening the analyses and improving the presentation. We particularly ask that you deposit all relevant data to public databases, perform additional analyses as suggested, and address all referee comments as thoroughly as possible with appropriate revisions. We hope that you will find the prioritized set of referee points to be useful when revising your study.

We therefore invite you to revise your manuscript taking into account all reviewer and editor comments. Please highlight all changes in the manuscript text file. At this stage we will need you to upload a copy of the manuscript in MS Word .docx or similar editable format.

*1) Include a "Response to referees" document detailing, point-by-point, how you addressed each

referee comment. If no action was taken to address a point, you must provide a compelling argument. This response will be sent back to the referees along with the revised manuscript.

*2) If you have not done so already please begin to revise your manuscript so that it conforms to our Article format instructions, available [here](http://www.nature.com/ng/authors/article_types/index.html). Refer also to any guidelines provided in this letter.

[redacted]

We hope to receive your revised manuscript within 3 to 6 months. If you cannot send it within this time, please let us know.

Sincerely,
Wei

Wei Li, PhD
Senior Editor

Nature Genetics
New York, NY 10004, USA
www.nature.com/ng

Reviewers' Comments:

Reviewer #1:
Remarks to the Author:
Overall evaluation

=====

This manuscript presents an amazing amount work about *Setaria* genomics and breeding genetics. 110 newly assembled genomes, more than 1,800 re-sequenced accessions in total, 68 traits assessed for a huge collection for up to 11 years... for sure an overwhelming amount of data, with which the authors did an excellent work and I image that it will be translated into many future agricultural improvements for *Setaria*. Nevertheless, I think that there are several parts that needs to be fixed before this manuscript can be published. One of them is associated with the manuscript structure. The results contain many statements that will be better fitted on the discussion. At the same time, the discussion is a summary of the results rather than a good opportunity to contextualize the results under several topics such as grass domestication (maybe it was included in previous articles, but I found missing here), *Setaria* genome evolution or even impact of newly developed graph-based pangenomic approaches on genetic breeding. I have also other minor concerns associated to assembly QC, interpretation of the gene-based pangenome without taking into account population structure, association between gene expression and PAVs and missing parts and details in the material and methods. Still, I think that this manuscript is a good example of how genomics is moving into the pan-genomic era and the advantages of it. Worthy of being published in this journal and for sure and enjoyable reading for a plant genomic scientist.

Point-By-Point Manuscript Evaluation:

=====

A. Summary of the key results

The manuscript titled "A graph-based genome and pan-genome variation of the model plant *Setaria*" describes the phylogenetic analysis of 1,844 *Setaria* samples as well as a simple population analysis using short read re-sequencing data. This analysis clustered the accession into seven groups (for associated to wild type accessions) and three associated to cultivated *S. viridis* accessions. This analysis drove to the selection of 110 accession for genome sequencing and assembling with long reads. Additionally, the sequencing depth was increased for three of these accessions to get a better assembly, adding also optical mapping data. The assemblies were evaluated with BUSCO, LAI and Illumina reads remapping, showing good values (e.g., BUSCO scores > 90%) for further analysis. The protein-coding genes of 113 genomes (110 produced in this study) were used to build a gene space pangenome with 73,663 gene families. 23.6% of the gene families were core genes.

These genome assemblies were used also to analyze the variants comparing them with the Yugu1

reference genome. More than 200,000 SVs were detected. The authors found that the number of SVs declines as they are closer to genic regions, inferring that it is related with their impact on the gene function. Several examples such as the OsSGT1 gene were cited. The authors also found that the genes with a PAVs (detected at least in 10% of the accessions) have a lower expression value than the ones without them. The PAVs were also analyzed in the context of the domestication *Setaria* using three different genome-wide selection signatures, finding 4,582 and 152 PAVs associated with domestication and improvement respectively.

This analysis was complemented with an experiment in which a RIL developed from a non-shattering cultivar crossed with a wild accession with the shattering trait. The authors identified three major QTLs associated with this trait. One of the QTLs drove to the proposal of SvLes1 as one of the genes associated with this trait. To reduce the number of the candidate genes, they developed NILs, pointing that the sh1 gene may be responsible for one of the QTLs.

The authors also used the resources generated to identify alleles associated with grain yield. A GWAS revealed 5 regions associated with this trait, of which the chromosome 3 had the stronger signal. A deeper analysis of the expression patterns of the genes under the GWAS of the chromosome 3 pointed the gene Seita.3G109700 as candidate gene regulating this trait. The authors performed a further validation with the overexpression of this gene, obtaining as expected plants with reduced grain width and increased grain length.

The final study presents a graph-based pangenome built using the PAVs identified in the previous study and its use with 1,844 short read re-sequenced accessions. 68 traits (226 phenotypes) were studied for these accessions in 13 different locations and 11 years. An interesting result that the authors present is that "36.9% of SVs were not in LD with their flanking SNPs (± 50 kb, $R^2 < 0.5$) which indicates the genetic information associated with SV would not be captured by SNP markers". The authors illustrate the utility of the pangenome with several traits for which they propose candidate genes such as: Apparent amylose content (Seita.4G022400 (GBSSI) gene); TGW and peduncle length (Seita.9G020100 (Gdh7 homolog) gene). Finally, the authors use the pangenome for genomic selection obtaining higher prediction accuracies than in other crops such as tomato.

B. Originality and significance: if not novel, please include reference

The manuscript is original. The authors present an incredible amount of data and results highlighting the advantages of the use of a pangenomic approach instead of a classical reference based. In this sense, the manuscript has a clear significance for the field of plant genomics and its application to plant breeding.

C. Data & methodology: validity of approach, quality of data, quality of presentation.

The used approaches are the state-of-the-art in genomics being valid for the studies presented in this manuscript. The quality of the data looks okay although there are some additional QC checks that could be performed on the genome assemblies (see section D of this revision). The quality of the presentation should be improved in some sections and figures (e.g., the result has some discussion, and the discussion is more a summary of the results than a well-developed discussion, there are some mistypes in a couple of figures...).

D. Appropriate use of statistics and treatment of uncertainties

Overall, the use of the statistics of the works presented in this manuscript is appropriate.

E. Conclusions: robustness, validity, reliability

The conclusions are well supported with the data presented in the manuscript, although there are many details missing in the material and methods as well as data not accessible in the supplementary data, so it is difficult to evaluate the reliability.

F. Suggested improvements: experiments, data for possible revision

a. The material and methods lack from numerous details, from how the plants were growth, the DNA extracted, and the different DNA sequencing libraries prepared to the specifications of the different reference genomes used in this work (e.g., genome ID...), some programs do not have their version number (e.g., ADMIXTURE, fastsimcoal2...). In the ADMIXTURE analysis and the demographic history inference, the authors did not supply the parameters used for fastsimcoal2 (e.g., number of coalescent simulations). There is not specification of where the RNA-Seq used in the annotation come from. The version of the other genomes' proteomes used for the annotation have not specified either.

b. There are minor details in the main text, tables and figures that should be revised. Some examples:

i. Figure 1c. "Number" has been misspelled as "nubmer".

ii. Suppl. Table 2 (and others later). Chromosome name is "scaffold_x" where I presume x is the chromosome number. This could be renamed to just the chromosome number (e.g., 1, 2...) or meaningful seqID (e.g., SevirxxxCy where xxx is the assembly version and y the chromosome number).

iii. Suppl. Table 4. "Predicated genome size" should be "Predicted genome size".

iv. Line 242: The word Seteria should be Setaria

c. The population genetic analysis results are not clear. Its reading drive to the perception that seven groups based on the phylogenetic tree. Then, reading the M&M section, it was stated that the optimal k value was seven based on the ADMIXTURE value (with k values ranging from 2 to 9). I recommend clarifying this in the result section. Additionally, although may not deliver additional results, it could be worthy to try higher K values (e.g., from 2 to 20), to see if there are some fractionations of the proposed groups. Finally, in this regard, it will be useful to have some pop. genetic metrics on these groups such as nucleotide diversity, heterozygosity (observed) and any other F-statistic value that can help to understand better some of the properties of these populations. It will be useful to have an extra column in the Suppl. Table 1 with the assignment of each of the accession to each of the described groups.

d. I found interesting the result from the demographic history inference where "Within the domesticated species, the C1 subgroup represents the first domesticated population, and subgroups C2 and C3 were subsequently established from C1, with evidence of gene flow from the wild W1 *S. viridis* subpopulation". Nevertheless, I couldn't find any continuation of this result in further analysis

neither in the final discussion. Did the authors find any other evidence of the gene flow, for example in the analysis of the pangenome? If so, did they find any gene associated with local adaptation? It will be interesting to see at least some other evidence of gene flow as well as some discussion about how it has contributed to the agronomical traits of these groups.

e. Would be possible to have a one sentence summarizing of the 110 sequenced accessions, how many are per genetic group (W1, W2, W3, W4, C1, C2 and C3). From Figure 2a, it looks like there is an under-representation of the groups W2 (0), W3 (1) and W4 (1). If so, how do the authors think that may impact the pan-genomic analysis? Would be better to consider these groups for the analysis of core and dispensable genes? (e.g., some soft-core gene present in 112 could be absent in the W4 genome... so it could be core if only W1, C1-3 genomes are considered).

f. For the evaluation of the genome assembly quality, I strongly recommend adding a Merqury analysis to estimate completeness, QV and levels of duplication based on the Kmer distribution. It could be interesting to classify the genome quality based on the Earth Biogenome Project standards (<https://www.earthbiogenome.org/assembly-standards>).

g. The authors describe that number of PAV in the ref-based pangenome as insertions, deletions, translocations, and inversions. Do they know how many of the insertions/deletions are associated to transposable elements? For example, do they know if the 124 bp deletion in the *Seita.1G213000* gene was produced by TE action? There are some examples in rice of how TE may be responsible for some of the deletions (Yan et al. 2022).

h. The analysis of the influence of the PAVs on the gene expression may not be well supported. If I understood correctly, the authors used an expression experiment on the "Yugu1" accession to infer a property in a whole population. I think that a stronger design could be to analyze the expression on several accessions and analyze the PAV only on those. Then perform an association analysis between gene expression (similar to a "phenotype") and PAV.

i. The authors wrote in the result section "The density of differentially expressed genes (DEGs) show significantly positive correlation with PAV density between both wild and cultivated foxtail millets (A10 versus Yugu1) and between landrace and modern cultivated accessions (Ci846 versus Yugu1)". Nevertheless, looking the Figure 6 the correlation values although they are positive, they are quite weak (low degree) ($r < 0.29$) although the showed p-values are significant. Under these results, I am not sure the conclusion "PAV may be involved in foxtail millet domestication and improvement by regulating associated gene expression" is well supported.

j. I am not familiar with the domestication syndrome in *Setaria*, but it will be interesting to have a list of traits associated to the domestication syndrome, pull out the GWAS peaks associated with them and see how many of them overlap with the domestication regions.

G. References: appropriate credit to previous work?

I think so (to my knowledge).

H. Clarity and context: lucidity of abstract/summary, appropriateness of abstract, introduction and conclusions

--

The manuscript is clear and easy to follow, although some parts of the results should be moved to the discussion and the late one, be improved from my point of view. The abstract is a little bit poor capturing some of the most relevant results of this manuscript from my point of view (e.g., from the sentence "Here, we de novo assembled 110 representative *Setaria* accessions from a worldwide collection of 1,844 varieties" jumps into "We establish the importance of pan-genome variation..." without describing any result of the pan-genome (e.g., core genes, number of PAVs..).

Manuscript revised by Aureliano Bombarely on Sept. 15th, 2022.

Reviewer #2:

Remarks to the Author:

This study newly resequenced 1004 *Setaria* accessions, de novo assembled 110 *Setaria* genomes and constructed a *Setaria* pan-genome. The authors investigated contributions of SV to foxtail millet domestication and improvement and found that SiGW3 with a 366-bp PAV regulates grain yield in foxtail millet. They also performed GWAS and GS studies for 226 phenotypes.

In general this work is very impressive and their findings facilitate foxtail millet breeding.

I have only few concerns about this study. My main concern is about the quality of de novo assembled genomes. The first, the authors obtained the genomes of three representative accessions with a mean contig N50 length >20 Mb, while they assembled the others with much shorter contig N50. Some accessions have been sequenced with very high depth data, e.g. W61 with 258.6X PacBio reads and 62.3X Illumina reads having contig N50 1.16Mb. Is the improvement of BioNano physical maps so significant? particularly at contig level... The second, the authors only used Illumina reads mapping, LAI and BUSCO to evaluate the accuracy of genome assembly. It may be not enough. For example, the authors should check and evaluate the synteny between assembled genomes (at least long contigs) and previously published chromosome-level genomes. The third, the accuracy of the SVs identified based on de novo assembled genomes should be checked manually (using PE reads, mapping depth, etc), especially for those SV cases mentioned in main text (other SVs may be randomly checked).

Other minor comments

1) In Fig. 1, the tree looks strange and the cultivated population likes root, please redraw it. Also, it is not necessary to show too many K numbers for structure analysis in a main figure. In addition, it may be better to integrate Figs 1 and 2 into one figure. Some information are repeated in these two figures.

2) L283, If the authors put 'parallel selection' in the section title, only broomcorn millet data is not enough and more data and evidences should be provided, otherwise I suggest removing the result about parallel evolution.

3) Although some SV cases related to domesticated genes are given in the manuscript, in my expectation, there should be a supplementary table listing identified SVs and their related genes with annotation of cloned genes (including homology to cloned genes from other cereal crops) and published QTLs (not just bioinformatics annotations listed in some tables now). In this way, readers

will get more effective information.

4) The methods should be more detailed and some parameters of software lack in most method sections.

Reviewer #3:

Remarks to the Author:

He et al. developed a graph-based genome for foxtail millet using genome assembly from 110 diverse accessions. Additionally, they identified SV involved in domestication and improvement, variants associated with various phenotypes. The manuscript is well structured. and will provide a resource for Setaria community and a path for upcoming pangenome manuscripts

1) Why was *Setaria viridis* included as part of graph-based genome development? It is a wild species and is about 5000-6000 years apart. At the genome level, both species are diverse and would add more ambiguity to the graph-based pan-genome.

2) Providing scripts for fastcoalsim for the models tested would be very helpful for the population genetic community, whether as a supplementary or a public repository.

Author Rebuttal to Initial comments

REFEREE 1 COMMENTS:

1. The material and methods lack from numerous details, from how the plants were growth, the DNA extracted, and the different DNA sequencing libraries prepared to the specifications of the different reference genomes used in this work (e.g., genome ID...), some programs do not have their version number (e.g., ADMIXTURE, fastsimcoal2...). In the ADMIXTURE analysis and the demographic history inference, the authors did not supply the parameters used for fastsimcoal2 (e.g., number of coalescent simulations). There is not specification of where the RNA-Seq used in the annotation come from. The version of the other genomes' proteomes used for the annotation have not specified either.

Response: Thanks for your suggestions. We have edited the Methods section to include the pertinent information. In our re-analysis of the population genomics, we realized that the fastsimcoal2 analysis has several issues associated with model specification; we have therefore removed it from the paper and redone the analyses using various other approaches (TREEMIX, Admixtools). We have deposited all related scripts/parameters more detailly at github: <https://github.com/qiangh06/Setaria-pan-genome>.

2. There are minor details in the main text, tables and figures that should be revised. Some examples:

i. Figure 1c. "Number" has been misspelled as "nubmer".

ii. Suppl. Table 2 (and others later). Chromosome name is "scaffold_x" where I presume x is the chromosome number. This could be renamed to just the chromosome number (e.g., 1, 2...) or meaningful seqID (e.g., SevirxxxCy where xxx is the assembly version and y the chromosome number).

- iii. Suppl. Table 4. “Predicated genome size” should be “Predicted genome size”.
- iv. Line 242: The word Seteria should be Setaria

Response: We have edited these sentences and double-checked all text carefully, making the necessary corrections

3. The population genetic analysis results are not clear. Its reading drive to the perception that seven groups based on the phylogenetic tree. Then, reading the M&M section, it was stated that the optimal k value was seven based on the ADMIXTURE value (with k values ranging from 2 to 9). I recommend clarifying this in the result section. Additionally, although may not deliver additional results, it could be worthy to try higher K values (e.g., from 2 to 20), to see if there are some fractionations of the proposed groups. Finally, in this regard, it will be useful to have some pop. genetic metrics on these groups such as nucleotide diversity, heterozygosity (observed) and any other F -statistic value that can help to understand better some of the properties of these populations. It will be useful to have an extra column in the Suppl. Table 1 with the assignment of each of the accession to each of the described groups.

Response: We apologize for the confusing description of the method. The method for inferring the optimal k value follows the approach used by the 3K-rice project, which was the minimal k value needed to separate all previous known groups in rice based on the ADMIXTURE result (Wang et al., 2018)¹. In this study, $k = 7$ was chosen because it displayed the minimal value of k to separate all previously known groups of *S. viridis*² (Mamidi et al., 2020) and *S. italica*³ (Jia et al., 2013) (W1, W2, W3, W4, C1 and C2) based on ADMIXTURE result. We have rephrased the text in the Results section Line 107 – Line 109. We also added the extra column about the subgroup information in **Supplementary Table S1**. As per your recommendation, we have also shown the groups that arise as we increase the k value to 20 (**Supplementary Fig2a**). As is now customary, these are for illustrative purposes and we focus our attention on $k = 7$ which is concordant with the subpopulation numbers derived from other analyses.

We also estimated nucleotide diversity, heterozygosity and pairwise F_{ST} analysis for these subpopulations and incorporated them into the paper (**Supplementary Fig. 2d**). The F_{ST} analysis showed that cultivated subgroups - C3 has the closest relations with wild subgroup-W1 (**Supplementary Fig. 2e**), which is consistent to the phylogeny of these subgroups. We added the results of these analyses as well in Line 121 – Line 124.

4. I found interesting the result from the demographic history inference where “Within the domesticated species, the C1 subgroup represents the first domesticated population, and subgroups C2 and C3 were subsequently established from C1, with evidence of gene flow from the wild W1 *S. viridis* subpopulation”. Nevertheless, I couldn’t find any continuation of this result in further analysis neither in the final discussion. Did the authors find any other evidence of the gene flow, for example in the analysis of the pangenome? If so, did they find any gene associated with local adaptation? It will be interesting to see at least some other evidence of gene flow as well as some discussion about how it has contributed to the agronomical traits of these groups.

Response: Thank you for this comment. The referee would like to see more analyses/discussion around the demographic inference we have laid out. S/he suggests looking for evidence of gene flow using the pangenome; however, we find that this will be difficult to conduct given how recent these populations have diverged and the possibility of incomplete lineage sorting (ILS) confounding the results. Instead, we have conducted three other population genomic analyses.

In looking over these analyses, we realized that our fastsimcoal2 analyses had issues with the model specification, and using it to do model selection was problematic. We therefore decided to completely rework the population genomic and demographic inference analyses using TREEMIX, Admixtools, and f4 analyses (to look at pattern of population divergence and identify gene flow). Our results in the three approaches (TREEMIX, admixture graph and f4) indicate there is no evidence of substantial gene flow into the cultivated millets from the W1 wild population. If any, there is some evidence of gene flow between cultivated to distant wild populations; however these do not appear to be substantial and does not impact our main focus on the domesticates. Finally, our analyses all point to an initial divergence between C3 vs. C1/C2, and a later divergence of C1 and C2.

Because of this, we have removed the fastsimcoal2 analyses and instead report the TREEMIX, admixture and f4 analyses with respect to branching order and gene flow. We have also extended the analyses by looking for genomic regions associated with adaptation in specific cultivated groups using Graph-aware Retrieval of Selective Sweeps (GROSS). In the latter case, we have compared the selection results to our PAV and GWAS analyses. We have incorporated all these results into the text (Lines 125-136).

5. Would be possible to have a one sentence summarizing of the 110 sequenced accessions, how many are per genetic group (W1, W2, W3, W4, C1, C2 and C3). From Figure 2a, it looks like there is an under-representation of the groups W2 (0), W3 (1) and W4 (1). If so, how the authors think that may impact the pan-genomic analysis? Would be better to consider these groups for the analysis of core and dispensable genes? (e.g., some soft-core gene present in 112 could be absent in the W4 genome... so it could be core if only W1, C1-3 genomes are considered).

Response: We added a sentence that summarized the distribution for the 110 accessions at lines 147-149.

Thank you for your question on the pan-genome core analysis with regards to a smaller number of accessions in subpopulations W2-W4. We agree this is an issue but would argue that this is not a major problem in our pan-genomic analysis. Theoretically, this is a systematic error that plagues all pan-genomic studies, wherein the “real pan-genome” is elusive due to the limited number of accessions in any study. Nevertheless, we feel that using W1, C1-C3 for our current pan-genome analysis is biologically meaningful, considering W1 is the evolutionary closest population with cultivated foxtail millet; they thus together represent a clear evolutionary lineage. Therefore our re-constructed gene-based pangenome of foxtail millet using all 80 cultivated accessions and 28 wild accessions from W1, and three released genomes represents an analysis of an evolutionary group that represents both the ancestral population of domesticated foxtail millet and all domesticated subpopulations. Thus, in our analysis, the pan-genome

was composed of 73,528 gene families, of which 23.8% were core genes, and 42.9% were soft core (those present in over 90% individuals, 100 to 110 accessions), 29.4% were dispensable (those present in 2-109 accessions), and 3.9% were private genes (**Fig. 3a**) We have added these results in our text (Lines 192-196).

6. For the evaluation of the genome assembly quality, I strongly recommend adding a Merqury analysis to estimate completeness, QV and levels of duplication based on the Kmer distribution. It could be interesting to classify the genome quality based on the Earth Biogenome Project standards (<https://www.earthbiogenome.org/assembly-standards>).

Response: Thank you for your suggestions. We have added the K-mer based analysis for completeness, QV and duplication levels for all assemblies in **Supplementary Tables S6**. We added the description in lines 164-166 and lines 172-176.

7. The authors describe that number of PAV in the ref-based pangenome as insertions, deletions, translocations, and inversions. Do they know how many of the insertions/deletions are associated to transposable elements? For example, do they know if the 124 bp deletion in the *Seita.1G213000* gene was produced by TE action? There are some examples in rice of how TE may be responsible for some of the deletions (Yan et al. 2022).

Response: Thank you for your suggestion. We have compared all PAVs with all annotated TEs in the assembled genomes, and add the description in lines 209 to 215. We find that most of the presence (72.3%, 59429) and absence (92.8%, 99477) variants overlapped with TEs, which are significantly higher than the genome-wide proportion of TEs (60.5%, $p < 0.001$) (**Supplementary Fig.4c**). These TE-associated PAVs were clustered in DNA transposon regions, and most of the breakpoints of those PAVs were close to the junction sites of TEs (**Supplementary Fig.4d-e**). These indicate that DNA transposons may have driven the formation of the most PAVs. We also identified 15,758 high confidence TE-derived PAVs, which co-located with single intact TEs coupled with target site duplications (TSD). We have added these in the text, and have deposited the PAV information at github: <https://github.com/qiangh06/Setaria-pan-genome/tree/main/DATA>).

We also checked the SVs we mentioned in the manuscript to determine whether they are high confidence TE-derived PAVs. However, only the 1.4 kb deletion in *Seita.2G276500*, the 627 bp deletion in *Seita.7G171000* and the 2.02 kb deletion in *Seita.8G046900*; and *Seita.8G047000* were co-located with fragmented TEs and none of them were co-located with single intact TEs. Meantime, we found 404 domPAV, 11 impPAV and 8 GWAS significant signals were co-located with single intact TEs (**Supplementary Tables 10-11 and 14**).

8. The analysis of the influence of the PAVs on the gene expression may not be well supported. If I understood correctly, the authors used an expression experiment on the “Yugu1” accession to infer a property in a whole population. I think that a stronger design could be to analyze the expression on

several accessions and analyze the PAV only on those. Then perform an association analysis between gene expression (similar to a “phenotype”) and PAV.

Response: We agree that to strengthen this point it would be best to get gene expression data in multiple varieties; however we feel this is a large undertaking and beyond the scope of this study. Nevertheless, we decided to rewrite the section (Lines 233-235) and also expanded the point in the Discussions, so it is clear what we are testing and to show that our analysis is applicable. We note that gene expression is believed to be under stabilizing selection, and highly expressed genes are under greater levels of stabilizing selection⁴⁻⁷ (Lye, Choi and Purugganan 2022; Lemos et al. 2005; Larracuente et al. 2008; Gout et al. 2010). This leads to the hypothesis that PAVs are less likely to be found in these highly expressed genes compared to lower expressed genes, and this is what we observe (**Supplementary Fig. 6a-b**). Even though we use only data from one variety, since we have a large number of genes tested ($n = 7,191$ genes) this gives us sufficient statistical power to test this prediction.

9. The authors wrote in the result section “The density of differentially expressed genes (DEGs) show significantly positive correlation with PAV density between both wild and cultivated foxtail millets (A10 versus Yugu1) and between landrace and modern cultivated accessions (Ci846 versus Yugu1)”. Nevertheless, looking the Figure 6 the correlation values although they are positive, they are quite weak (low degree) ($r < 0.29$) although the showed p-values are significant. Under these results, I am not sure the conclusion “PAV may be involved in foxtail millet domestication and improvement by regulating associated gene expression” is well supported.

Response: We actually find that the correlations are moderate, not weak; between wild and cultivated accessions R ranging from 0.42 to 0.64 and between landrace and modern cultivated accessions ranging from 0.41 to 0.48 depending on tissue (**Supplementary Figs. 6d-e**) (We have changed R^2 to R in **Supplementary Figs. 6d-e** this time). Moreover, these correlations are all highly significant ($p < 2.2 \times 10^{-16}$). However, we understand the concern of connecting PAVs directly with crop domestication and improvement based solely on this analysis. We have therefore rewritten this section in lines 243 - 250 to just state that this correlation is significant, and indicate its role on domestication and improvement as suggestive (rather than conclusive).

10. I am not familiar with the domestication syndrome is *Setaria*, but it will be interesting to have a list of traits associated to the domestication syndrome, pull out the GWAS peaks associated with them and see how many of them overlap with the domestication regions.

Response: The only traits that are unambiguously domestication traits in our data set are non-shattering, increased grain weight and size (for some of the other traits, it is unclear if they are domestication or improvement traits). In our paper we have focused on these two key domestication syndrome traits – seed non-shattering and increased grain yield. To make this clearer, we have rewritten the section on domPAVs, impPAVs, shattering and *SiGW3*. We have integrated the analyses of domPAVs and

impPAVs into one section, and seed non-shattering and grain yield into a separate section (Lines 286-351). Together, we make clear that we systematically analyzed these two key domestication traits. For the GWAS analysis, only grain weight and grain width were examined as likely domestication traits, and all of the GWAS hits in our analyses all span domPAVS; we have added this in the Results (Lines 400-402). Note that our goal is to show that PAVs contribute to domestication, which we hope the rewritten sections make clear.

11. The manuscript is clear and easy to follow, although some parts of the results should be moved to the discussion and the late one, be improved from my point of view. The abstract is a little bit poor capturing some of the most relevant results of this manuscript from my point of view (e.g., from the sentence “Here, we de novo assembled 110 representative *Setaria* accessions from a worldwide collection of 1,844 varieties” jumps into “We establish the importance of pan-genome variation...” without describing any result of the pan-genome (e.g., core genes, number of PAVs...).

Response: Thank you for your suggestions. We have rewritten the abstract to highlight our results (although the word limit constrains what we can put in the abstract). We have rewritten the Discussion so that it is less of a recap of the Results, but instead provides a broader context to some of the key points – the importance of pan-genome variation, and the critical roles that the availability of these unique genome resources will play in advancing genetic analysis and breeding of this model C4 crop species. We also accepted the suggestion of the referee and added a short paragraph on some of the implications of our study on foxtail millet evolution. Finally, we also took the reviewer’s suggestion and moved some parts from the Results to the Discussion.

REFEREE 2 COMMENTS:

1. I have only few concerns about this study. My main concern is about the quality of de novo assembled genomes. The first, the authors obtained the genomes of three representative accessions with a mean contig N50 length >20 Mb, while they assembled the others with much shorter contig N50. Some accessions have been sequenced with very high depth data, e.g. W61 with 258.6X PacBio reads and 62.3X Illumina reads having contig N50 1.16Mb. Is the improvement of BioNano physical maps so significant? particularly at contig level... The second, the authors only used Illumina reads mapping, LAI and BUSCO to evaluate the accuracy of genome assembly. It may be not enough. For example, the authors should check and evaluate the synteny between assembled genomes (at least long contigs) and previously published chromosome-level genomes. The third, the accuracy of the SVs identified based on de novo assembled genomes should be checked manually (using PE reads, mapping depth, etc), especially for those SV cases mentioned in main text (other SVs may be randomly checked).

Response: Thank you for your constructive questions/suggestions. We address these in turn below:.

A. According to your and reviewer #1's suggestions, we have added the Merqury analysis to estimate completeness, quality value (QV) and false duplication levels based on the Kmer distribution, to evaluate our assemblies. As we expected, the genome quality of modern cultivated accessions (completeness: $97.59\% \pm 2.02\%$, QV: 39.36 ± 1.78 , duplication: $2.55\% \pm 1.16\%$) are higher than wild accessions (completeness: $91.34\% \pm 6.05\%$, QV: 30.52 ± 6.89 , duplication: $4.34\% \pm 2.48\%$). Most of the assemblies have over 95% completeness (70 cultivated accessions and 8 wild accessions), with only 5 wild accessions having relatively low completeness (<85%). This is mainly due to the difference in heterozygosity of these accessions. However, we think these genomes are of sufficiently high quality for our study, and to construct a graph-based genome and examine the pan-genome variation of *Setaria*.

We think there are three reasons why our three representative accessions have longer contig N50s compared to other accessions: (i) these accessions all have very low heterozygosities (Yugu18: 0.11%, Ci846: 0.06%, Me34V: 0.08%); (ii) We generated more sequencing data for these three accessions than the others. These three accessions had $118.3 \pm 4.0X$ long read sequence coverage, while most of the rest of the accessions (93 accessions) had $79.9 \pm 18.2X$ coverage. Only 14 accessions - 12 of them having higher heterozygosity than these three accessions - had relatively higher sequencing depth (119.3X~258.6X) (**Supplementary Table 5**). We compared the correlation between long read sequencing depth and the length of contig N50 for the accessions. The contig N50 and sequencing depth showed significant positive correlation ($R=0.504$). Only two accessions W61 and Liushabai had higher sequencing depth (258.6X and 242.1X) but lower contig N50s (1.16M and 1.27M). (iii) Finally, Bionano physical maps helped in joining position detection and error correction for the original contigs, which could have improved the final contig accuracy and the continuation degree (contig N50 increased) using the HERA software.

B. We have checked the synteny between all our assemblies and the foxtail millet reference genome "Yugu1" (v2.2) (**Supplementary Table 6**). A higher degree of genome synteny is retained between cultivated accessions (0.94 ± 0.02) compared to wild accessions (0.88 ± 0.02), consistent with evolutionary relationships. Wild accession Q13 has the lowest synteny ratio (~75.03%) and K-mer completeness (72.42%), due to its very high heterozygosity (0.96%).

C. Last time, we manually checked the SVs mentioned in the manuscript with long read sequence data (**Supplementary Fig.5a-d**). This time we manually checked for these positions by both long-read and PE short read data and updated the **Supplementary Fig. 5a-d**. Moreover, we randomly selected 27 PAVs distributed on 9 chromosomes and re-checked by Pacbio long-read and PE short-read data (**Response Fig. 1**). All of these SVs were supported by both long-read and short-read alignments.

Response Fig. 1 Physical position and reads alignment for manually checked 27 random selected SVs.

Some minor comments:

2. In Fig. 1, the tree looks strange and the cultivated population likes root, please redraw it. Also, it is not necessary to show too many K numbers for structure analysis in a main figure. In addition, it may be better to integrate Figs 1 and 2 into one figure. Some information are repeated in these two figures.

Response: Thank you for your suggestions. We have redrawn the phylogenetic tree to unrooted type, which we hope makes it clearer (**Fig. 1a**). As to the population structure analysis, showing more K patterns allows us to explore main patterns of differentiation among cultivated foxtail millet as K increases. Thus, we prefer to keep the STRUCTURE results in **Fig. 1** as we had in the original submission, but we also increased K to 20 in **Supplementary Fig. 2a**.

In our study, **Fig. 1** gives an overview of the population structure, phylogenetic relations, and geographic distributions for *Setaria* (foxtail millet and green foxtail), which are focused on the evolution of *Setaria*, while **Fig. 2** mainly describes the diversity of the representative 110 assembled accessions. Considering the manuscript structure, we hope to keep these two figures as before. We thank you for your understanding.

3. L283, If the authors put ‘parallel selection’ in the section title, only broomcorn millet data is not enough and more data and evidences should be provided, otherwise I suggest removing the result about parallel evolution.

Response: We have rewritten this section and deleted the mention of parallel selection in lines 348-351; we simply note in the end that there is evidence in 3 lineages (foxtail millet, broomcorn millet and rice) that homologues of the same gene are responsible for the same trait.

4. Although some SV cases related to domesticated genes are given in the manuscript, in my expectation, there should be a supplementary table listing identified SVs and their related genes with annotation of cloned genes (including homology to cloned genes from other cereal crops) and published QTLs (not just bioinformatics annotations listed in some tables now). In this way, readers will get more effective information.

Response: Thank you for your suggestions. We have added more annotation with cloned genes from 27 published papers of foxtail millet, homology information to rice cloned genes from www.ricedata.cn, and QTLs collected from 14 published papers for 31 traits for foxtail millet in **Supplementary Table 10** and **Supplementary Table 11**.

5. The methods should be more detailed and some parameters of software lack in most method sections.

Response: We have added more detailed information in the Methods sections. Most of the analyses were performed with default parameters in the software, and we deposited the corresponding scripts and parameters for genome assembly, genome annotation, graph-genome construction, phylogenetics and demographic inference analyses on github: <https://github.com/qiangh06/Setaria-pan-genome>.

REFEREE 3 COMMENTS:

1. Why was *Setaria viridis* included as part of graph-based genome development? It is a wild species and is about 5000-6000 years apart. At the genome level, both species are diverse and would add more ambiguity to the graph-based pan-genome.

Response: Thank you for your comment. The main purpose of pan genome or graph-based genome study, is to capture the entire genomic repertoire in a species or closely-related species group, which could encode for the entire range of functional characteristics of that group of organisms⁸ (Vernikos et al.,

2015). We should point out that while domesticated foxtail millet may have diverged from wild *S. viridis* thousands of years ago, in genetic terms there still remains shared allelic variation between *S. viridis* and *S. italica*, largely from incomplete lineage sorting.

The issue with crops is also practical. Crop evolution and domestication has dramatically reduced genetic diversity, sometimes accompanied by gene loss. This reduced diversity or lost genes may play important roles in biotic/abiotic stress, flavor or even yield, and retrieving these functionally important genes/diversity may be significant for modern cultivar improvement. As the progenitor of cultivated foxtail millet, *S. viridis* has proved to be great resource for discovering agronomically valuable loci² and for foxtail millet improvement. Here, we used 187,904 structure variants (85,101 from 36 assembled wild genomes) for graph genome development (**Response Fig. 2**). When we used this graph-based genome as a reference to genotyping 1,211 cultivated accessions, we find that 74.6% (63,446) SVs from wild accessions, can also be detected in cultivated accessions. This suggests we cannot capture these 64,446 SVs, if we only use assembled cultivated accessions for graph genome construction. In our study, we think using both cultivated and closely-related wild species does not introduce ambiguity in graph-based pan-genome reconstruction, but instead significantly increases the genotyping capacity for both cultivated and wild foxtail millet.

Response Fig. 2 The origin of structure variations for *Setaria* graph-based genome development.

2. Providing scripts for fastcoalsim for the models tested would be very helpful for the population genetic community, whether as a supplementary or a public repository.

Response: Thank you for your suggestion. As we indicated in our response to Referee 1, we have actually replaced the fastsimcoal analyses with a battery of other tools (TREEMIX, Admixtools, GRoSS) for our evolutionary analyses. We have deposited the corresponding scripts on github:

<https://github.com/qiangh06/Setaria-pan-genome/tree/main/Population%20genomic%20and%20Demographic%20inference>.

Reference

1. Wang, W. *et al.* Genomic variation in 3,010 diverse accessions of Asian cultivated rice. *Nature* **557**, 43–49 (2018).

2. Mamidi, S. *et al.* A genome resource for green millet *Setaria viridis* enables discovery of agronomically valuable loci. *Nature Biotechnology* **38**, 1203–1210 (2020).
3. Jia, G. *et al.* A haplotype map of genomic variations and genome-wide association studies of agronomic traits in foxtail millet (*Setaria italica*). *Nature Genetics* **45**, 957–961 (2013).
4. Lye, Z., Choi, J. Y. & Purugganan, M. D. Deleterious Mutations and the Rare Allele Burden on Rice Gene Expression. *Molecular Biology and Evolution* **39**, msac193 (2022).
5. Larracuente, A. M. *et al.* Evolution of protein-coding genes in *Drosophila*. *Trends in Genetics* **24**, 114–123 (2008).
6. Gout, J.-F., Kahn, D., Duret, L. & Consortium, P. P.-G. The Relationship among Gene Expression, the Evolution of Gene Dosage, and the Rate of Protein Evolution. *PLOS Genetics* **6**, e1000944 (2010).
7. Lemos, B., Bettencourt, B. R., Meiklejohn, C. D. & Hartl, D. L. Evolution of Proteins and Gene Expression Levels are Coupled in *Drosophila* and are Independently Associated with mRNA Abundance, Protein Length, and Number of Protein-Protein Interactions. *Molecular Biology and Evolution* **22**, 1345–1354 (2005).
8. Vernikos, G., Medini, D., Riley, D. R. & Tettelin, H. Ten years of pan-genome analyses. *Current Opinion in Microbiology* **23**, 148–154 (2015).

Decision Letter, first revision:

: Our ref: NG-A60659R

13th Feb 2023

Dear Dr. Purugganan,

Thank you for submitting your revised manuscript "A graph-based genome and pan-genome variation of the model plant *Setaria*" (NG-A60659R). It has now been seen by the original referees and their comments are below. The reviewers find that the paper has improved in revision, and therefore we'll be happy in principle to publish it in *Nature Genetics*, pending minor revisions to comply with our editorial and formatting guidelines.

Sincerely,

Wei Li, PhD
Senior Editor
Nature Genetics

Reviewer #1 (Remarks to the Author):

I would like to thank the authors for the detailed answers to my comments and concerns. I think that the new version of the manuscript resolves successfully the different concerns risen during my first revision. I do not have any further comments but congratulate the authors for the quality of the work presented in this manuscript.

Reviewed by Aureliano Bombarely on Jan. 10th, 2023

Reviewer #2 (Remarks to the Author):

I am satisfied that the authors have appropriately addressed my comments on the manuscript. I feel that this revised manuscript is of high quality and would be of interest to the readers of Nature Genetics.

Reviewer #3 (Remarks to the Author):

The authors included all the suggestions and the manuscript is good in its current form

Final Decision Letter:

8th May 2023

Dear Dr. Purugganan,

I am delighted to say that your manuscript "A graph-based genome and pan-genome variation of the model plant *Setaria*" has been accepted for publication in an upcoming issue of Nature Genetics.

Your paper will be published online after we receive your corrections and will appear in print in the next available issue. You can find out your date of online publication by contacting the Nature Press Office (press@nature.com) after sending your e-proof corrections. Now is the time to inform your Public Relations or Press Office about your paper, as they might be interested in promoting its publication. This will allow them time to prepare an accurate and satisfactory press release. Include your manuscript tracking number (NG-A60659R1) and the name of the journal, which they will need when they contact our Press Office.

Please note that *Nature Genetics* is a Transformative Journal (TJ). Authors may publish their research with us through the traditional subscription access route or make their paper immediately open access through payment of an article-processing charge (APC). Authors will not be required to make a final decision about access to their article until it has been accepted. [Find out more about Transformative Journals](https://www.springernature.com/gp/open-research/transformative-journals)

Authors may need to take specific actions to achieve [compliance](https://www.springernature.com/gp/open-research/funding/policy-compliance-faqs) with funder and institutional open access mandates. If your research is supported by a funder that requires immediate open access (e.g. according to [Plan S principles](https://www.springernature.com/gp/open-research/plan-s-compliance)) then you should select the gold OA route, and we will direct you to the compliant route where possible. For authors selecting the subscription publication route, the journal's standard licensing terms will need to be accepted, including [self-archiving-and-license-to-publish](https://www.nature.com/nature-portfolio/editorial-policies/self-archiving-and-license-to-publish). Those licensing terms will supersede any other terms that the author or any third party may assert apply to any version of the manuscript.

Please note that Nature Portfolio offers an immediate open access option only for papers that were first submitted after 1 January, 2021.

If you have not already done so, we invite you to upload the step-by-step protocols used in this manuscript to the Protocols Exchange, part of our on-line web resource, natureprotocols.com. If you complete the upload by the time you receive your manuscript proofs, we can insert links in your article that lead directly to the protocol details. Your protocol will be made freely available upon publication of your paper. By participating in natureprotocols.com, you are enabling researchers to more readily reproduce or adapt the methodology you use. [Natureprotocols.com](https://natureprotocols.com) is fully searchable, providing your protocols and paper with increased utility and visibility. Please submit your protocol to <https://protocolexchange.researchsquare.com/>. After entering your [nature.com](https://www.nature.com) username and password you will need to enter your manuscript number (NG-A60659R1). Further information can be found at <https://www.nature.com/nature-portfolio/editorial-policies/reporting-standards#protocols>

Sincerely,
Wei

Wei Li, PhD
Senior Editor
Nature Genetics
New York, NY 10004, USA
www.nature.com/ng